# RETROSPECTIVE LEARNING FROM INTERACTIONS

## ABSTRACT

Multi-turn interactions between large language models (LLMs) and users naturally include implicit feedback signals. If an LLM responds in an unexpected way to an instruction, the user is likely to signal it by rephrasing the request, expressing frustration, or pivoting to an alternative task. Such signals are task-independent and occupy a relatively constrained subspace of language, allowing the LLM to identify them even if it fails on the actual task. This creates an avenue for continually learning from interactions without additional annotations. We introduce RESPECT, a method to learn from such signals in past interactions via retrospection. We deploy RESPECT in a new multimodal interaction scenario, where humans instruct a multimodal LLM to solve an abstract reasoning task with a combinatorial solution space. Through thousands of interactions with humans, we show how RESPECT gradually improves task completion rate from 31% to 82%, all without any external annotation.

## 1 INTRODUCTION

Language models (LMs) often engage in multi-turn interactions with human users. Similar to human-human interactions, these interactions are naturally rich with implicit learning signals. If the LM fails to respond appropriately, the user is likely to follow with an expression of frustration, a rephrase of their intent, or maybe even completely pivot what they ask for. Similarly, if the LM does well, the user may express approval or simply continue to their next objective. Such responses can inform the LM of its performance, thereby creating an opportunity to learn through retrospection.

We study the efficacy of such signals, and how they can lead to a system that improves over time. We introduce RESPECT, a simple approach to learn from signals the model itself derives about its own past actions through retrospection of past interactions with human users. We experiment with RESPECT by deploying a multimodal LLM (MLLM) in MULTIREF, a new multi-turn grounded

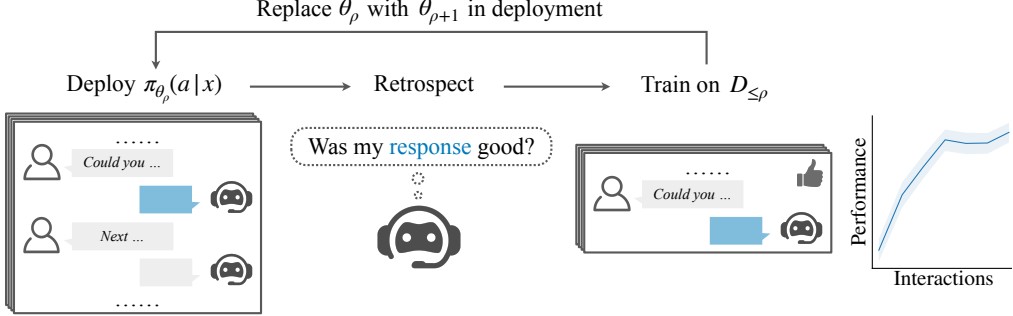

Figure 1: Learning via RESPECT. We deploy an MLLM policy $\pi_{\theta_\rho}(a|x)$ in rounds $\rho$, to interact with users in multi-turn interactions. Following each round, the LLM reasons retrospectively about each of its actions (highlighted in blue) to decode feedback given the interaction context, including follow up utterances. The decoded feedback can be positive (thumbs up as illustrated), negative or neutral. After each round, the model is retrained using all data aggregated so far $D_{\leq\rho}$. The MLLM improves over time without any external annotations. The plot on the right shows the performance curve in our experiments – the MLLM improves from 31% to 82% task completion rate over six rounds.

interaction scenario. MULTIREF is a generalization of reference games (Rosenberg & Cohen, 1964), and requires models to display complex abstract reasoning, and humans to gradually instruct models to accomplish sequences of goals to complete their tasks.

The key insight underlying RESPECT is that conversational implicit feedback signals occupy a relatively constrained subspace of natural language. Such signals can include direct approvals (e.g., *great!*) or signs of frustration (e.g., *not again*), and also more subtle cues, such as when the user rephrases their request. Critically, it is relatively simple to disentangle them from task performance. A human can easily figure out from such cues if they do well or not, even if they have little understanding about what they are asked for. It is this constrained nature that makes reasoning about such signals to be within the capacities of large language models (LLMs), even if they fail at the task at hand.

RESPECT utilizes this signal in a process where the model interacts with humans, and after interaction decodes feedback for each of its actions from the interaction context including the follow up utterances. Figure 1 illustrates this process. The model interacts with humans to accomplish tasks, retrospectively examines its own past interactions, and then re-trains. This process progresses in rounds, alternating between interaction and training, with the model improving over time. Critically, unlike common recipes for training from human feedback, RESPECT does not require any external annotation (Ouyang et al., 2022, RLHF) or even soliciting feedback from the users themselves (Suhr & Artzi, 2023).

We deploy RESPECT in MULTIREF over multiple rounds of grounded interactions with human use and re-training. We use IDEFICS2-8B (Laurençon et al., 2024) as our MLLM, and experiment with multiple learning methods, including supervised learning, REINFORCE-style policy gradient (Williams, 1992; Kojima et al., 2021), and KTO (Ethayarajh et al., 2024). Across our experiments, we observe that IDEFICS2-8B effectively decodes feedback, even as it initially performs poorly in the same interactions. In our longest running experiment, we observe model task completion rate improves from 31% to 82%. Our code, data, and models will be released upon publication.

## 2 TECHNICAL OVERVIEW AND NOTATION

We conduct continual learning[1] studies by deploying our approach in MULTIREF, a new multi-turn grounded interaction scenario (Section 3). Overall, the study progresses in rounds, where the MLLM policy is first deployed to interact with users and complete tasks, and the interactions are then used to re-train the policy. Our study involves multiple rounds, and our goal is to observe and evaluate the long-term dynamics of the process. This includes the robustness of our award decoding and training methods to the changing distribution of the data likely to be seen in an adaptive system in the wild. Section 3 describes our interaction scenario in detail, and Section 4 our learning method. First, we outline our problem of interest and its notation in abstract terms.

**Task Notation** The policy's task is to respond effectively to human utterances given in conversational context. Formally, let $\pi(a_t|x_t)$ be the policy that controls the listener behavior, with $a_t$ an action string that represents the model response and $x_t$ being the context on which the policy is conditioned, both at turn $t$ in the interaction. The context includes the instruction history up to and excluding turn $t$, including current (i.e., at turn $t-1$) and past speaker utterances, as well as any other relevant context in which the interaction takes place. As our learning progresses in rounds, we denote $\theta_\rho$ as the model parameters in round $\rho$, and $\pi_{\theta_\rho}$ as the parameterized policy.

**Learning and Deployment** We study a continual learning setup, where the learning signal is acquired from interactions of the deployed model with human speakers. Our study progresses in rounds (Figure 1). Each round $\rho$ includes a deployment, followed by training. During deployment at round $\rho$, the model $\pi_{\theta_\rho}$ interacts with users. For each model action $\hat{a}_t \sim \pi_{\theta_\rho}(a|x_t)$, we record a tuple $(x_t, \hat{a}_t, p_t, \bar{f}_t)$, where $x_t$ is the context given to the model at time $t$ to predict action $\hat{a}_t$, $p_t$ is the probability of $\hat{a}_t$ at the time of prediction, and $\bar{f}_t$ is the remainder of the interaction following $\hat{a}_t$. Critically, these interaction tuples contain no explicit feedback. We compute the implicit feedback $\hat{\gamma}_t$ using a feedback decoder $\phi(x_t, \hat{a}_t, \bar{f}_t)$, to obtain tuples $(x_t, \hat{a}_t, \hat{\gamma}_t, p_t)$. We experiment with multiple learning objectives using this feedback: supervised learning (SFT), policy gradient, and KTO.

---

[1]We define continual learning as the model improving over time on its task through interaction with human users. The term continual learning is used broadly for other purposes such as domain adaptation.

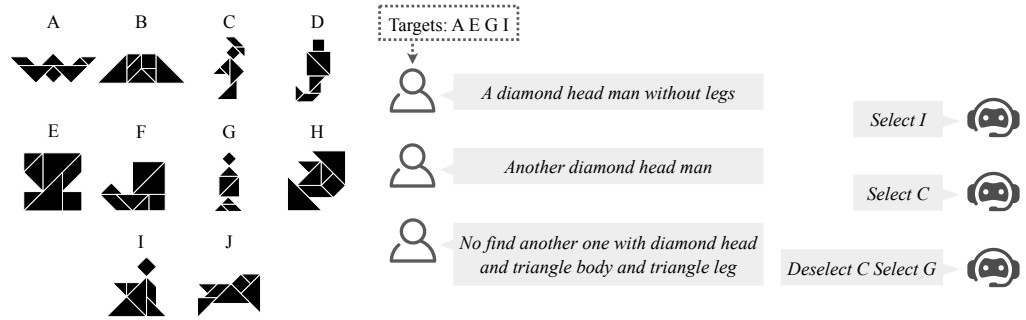

Figure 2: The interaction scenario we use in our experiments. MULTIREF is a multi-turn reference game. A speaker and a listener both observe a shared set of tangram shapes, but in different order. The goal of the speaker is to describe a subset of targets for the listener to select. Because the target requires multiple abstract shapes, humans often communicate the targets gradually over multiple turns. As an interaction progresses naturally, the speaker produces implicit feedback signals that validate or reject the listener's actions.

**Evaluation**   We measure the quality of the listener model $\pi_{\theta_\rho}(a_t|x_t)$ at each round $\rho$ primarily by interaction success rates from live human-bot deployments. The same interactions are used to train the model for the next round. We track various characteristics of model behavior, such as number of turns per interaction as an efficiency measure. We also do post-hoc annotation of a subset of the interactions to measure utterance-level policy success rate and feedback decoder accuracy.

## 3   MULTIREF: A MULTI-TURN GROUNDED INTERACTION SCENARIO

Key to our study is that tasks are relayed gradually across multiple turns, as commonly happens in human interactions. We create MULTIREF, a conversational interaction scenario where two partners, a *speaker* and a *listener*, coordinate on the selection of a set of items. In our studies, the speaker is always a human, and the listener is a model.

MULTIREF generalizes the commonly studied reference game scenario. Its design and our choice of stimuli are grounded in existing work from both cognitive science and computational language modeling (Rosenberg & Cohen, 1964; Clark & Wilkes-Gibbs, 1986; Schober & Clark, 1989; Goodman & Frank, 2016). Figure 2 illustrates the scenario. Both partners observe a shared set of images, but in different order. The speaker is given a subset of the images as targets, with the goal of communicating the targets to the listener, so the latter selects the exact subset. Only the speaker can write messages, and only the listener can select or deselect images. The interaction concludes successfully once all and only targets are selected, or fails if the partners run out of turns, 20 in our studies.

The interaction progresses in turns $t$, alternating between speaker and listener turns. At each speaker turn, they provide a single unrestricted natural language utterance. It may direct the listener to select one or more items, ask to deselect items if the listener previously made a mistake, or include whatever other content they desire. This utterance as well as the history of the interaction, the set of images, and their selection status compose the context $x_t$ for the following model turn at turn $t$. The follower responds with an action $a_t$, which includes one or more image selects or deselects according to their understanding of the speaker intention. The action space consists of all possible legal sequences of the form `Deselect E select F` or `Select D G` assuming images are code-named alphabetically.

The motivation behind MULTIREF is to create a task-oriented scenario that is both accessible to non-expert humans and encourages constructing a solution in multiple turns, thereby creating multi-turn interactions and eliciting the learning signals we aim to study. The rules of the interaction are simple: the speaker describes targets to select, and the listener selects what the speaker is referring to. This makes MULTIREF easily accessible to crowdsourcing workers. At the same time, the solution the speaker communicates to the listener is relatively complex, because of the enormous solution space: Consider a conventional reference games, where the goal is to select a single image. The number of possible solutions is the number of images in the context. In MULTIREF, the goal is to select

a subset of unknown size, so the combinatorial solution space the listener faces is exponential in the number of images. Meanwhile, the solution is decomposable, and the speaker can comment on the impact immediately after the listener's action (unlike Haber et al. (2019, PhotoBook)), creating natural opportunities to decompose the solution and for implicit incremental feedback to appear.

Key to making MULTIREF work well is the choice of images. We use tangram shapes from the diverse KILOGRAM dataset (Ji et al., 2022). Tangrams are abstract shapes that are designed to elicit common concepts in humans. This abstractness often leads to ambiguous descriptions open to interpretation, e.g., Shape A in Figure 2 can be described as a *bat*, a *lowercase w*, or even a *star wars star fighter*. We select tangrams because they naturally provide an ambiguous and challenging stimuli for human interaction (Clark & Wilkes-Gibbs, 1986; Schober & Clark, 1989; Fox Tree, 1999; Hawkins et al., 2020b), thereby leading to highly diverse language. They also remain challenging for contemporary MLLMs to reason about (Ji et al., 2022), leaving significant room for learning.

The free-form natural language human speakers produce in MULTIREF is very diverse, and balances between competing pressures. First, it often requires complex pragmatic reasoning (Clark & Wilkes-Gibbs, 1986; Schober & Clark, 1989; Horton & Gerrig, 2002), because of the abstractness of tangrams. This is compounded by how the combinatorial solution space drives humans to balance between relaying as much information as possible, and relaying clear objectives to make gradual progress. This is a balance between two Gricean maxims: quantity and manner.[2] Speakers may or may not include explicit feedback such as *good*, or *deselect the last one*; the speaker may describe more than one target in a single utterance, for example, *select two men*; speakers may refer to previous selections without directly describing targets, for example, *the other one*, or *try again*. In combination with the abstract stimuli tangrams provide, this creates a challenging reasoning problem for the listener model.

MULTIREF is not designed to increase complexity in arbitrary ways, but to provide an environment for humans to naturally expose core aspects of human communication. At the same time, the scenario is both controlled and scoped, allowing for easy measurement of task completion and progress, as well as making learning feasible with relatively limited data. This makes MULTIREF particularly suitable to research in academia or other low-resource settings.

## 4    RESPECT: RETROSPECTIVE LEARNING FROM PAST INTERACTIONS

RESPECT has two components: decoding implicit feedback from past interactions (*retrospection*) and learning from the decoded feedback signals (*learning*). We deploy RESPECT in an iterative continual learning scenario, where each round includes both steps. This deployment allows us to observe the dynamics of RESPECT over time. However, the method itself is not limited to continual learning, and can be applied a single time as well.

The goal of RESPECT is to re-estimate the parameters of a model given interactions that were collected by the model itself, or previous versions of it. We assume access to a raw dataset $D^{\text{raw}} = \{(x^{(i)}, \hat{a}^{(i)}, p^{(i)}, \bar{f}^{(i)})\}_{i=1}^{N}$, where $x^{(i)}$ is the policy context, $\hat{a}^{(i)}$ is the predicted action, $p^{(i)}$ is the probability of this action, and $\bar{f}^{(i)}$ is the remainder of the interaction following $\hat{a}^{(i)}$.[3] In our continual learning setup, $D^{\text{raw}}$ is a union of all data collected from past rounds.

The feedback decoder $\phi$ computes a categorical feedback $\hat{\gamma}^{(i)} \in \{\texttt{positive, neutral, negative}\}$ for each action $\hat{a}^{(i)}$ holistically based its context $x^{(i)}$, action taken $\hat{a}^{(i)}$, follow up utterances $\bar{f}^{(i)}$.[4] This process transforms $D^{\text{raw}}$ to $D = \{(x^{(i)}, \hat{a}^{(i)}, p^{(i)}, \hat{\gamma}^{(i)})\}_{i=1}^{N}$. We use this dataset for training.

### 4.1    DECODING IMPLICIT FEEDBACK THROUGH RETROSPECTION

We implement the feedback decoder $\phi$ by prompting the model to analyze past interaction tuples $(x, \hat{a}, p, \bar{f})$ to compute feedback $\hat{\gamma} = \phi(x, \hat{a}, \bar{f})$. The goal is a process where the model bootstraps

---

[2]Grice's maxim of quantity: one tries to be as informative as one possibly can, and gives as much information as is needed, and no more; Grice's maxim of manner: one tries to be as clear, as brief, and as orderly as one can in what one says, and where one avoids obscurity and ambiguity. (Grice, 1975)

[3]For simplicity of notation, we omit the turn step in this section.

[4]We do not compute feedback for the last action in each interaction because there is not followup interaction. For simplicity, $D^{\text{raw}}$ does not include them.

---

**Feedback Decoder Prompt**

User: Please carefully read the following conversation and answer: Is the very last utterance from the speaker positive or negative positive, neutral, or negative feedback? Often negative feedback include corrections and keywords like no, not, undo, don't, with generally negative sentiment, while positive feedback often includes good, yes, correct, okay, or simply move on to the next stage. Lean towards negative if it sounds neutral.
(start of the conversation)

    Listener: Deselect F select G

    Speaker: yes, pick the thin person with a triangle head

    Listener: Select A . . . . . . . . . . . . . . . . . . . . . . . . . . . . . . . . . . . . . . . *(Action to focus on)*

    Speaker: yes, pick the house with chimney . . . . . . . . . . . . . . . . . . . . . . . *(Feedback)*

(end of the conversation)
Answer a single word, Positive, or Negative Positive, Neutral or Negative.
Assistant: **Positive**

---

Figure 3: The prompt used to decode feedback from past interactions. The figure combines the prompts for both binary and ternary feedback decoding. The parts that belong to the binary case only are colored green, while parts that belong the ternary case are colored orange. The verbal **feedback generated by the model** is in bold. Additional *comments for readability* are in magenta italics.

from its own interactions. Our hypothesis is that LLMs have the ability to reason about the relatively constrained space of implicit signals, even if they fail at the task. We show this empirically in our experiments. Critically, this process does not rely on a stronger LLM for critique or on past interactions created by other LLMs. Figure 3 shows the decoder prompt. We experiment with binary or ternary feedback. Ternary adds `neutral` on top of the `positive` and `negative` binary options. The feedback decoder is designed to identify general linguistic cues, and not for the specific task we study. We assume no access to any auxiliary annotation or privileged information (e.g., not inferring based on whether the policy selects a ground truth target in a turn, or whether an entire interaction ends early), although they are likely to be useful signals as explored in Pang et al. (2023).

### 4.2 LEARNING

The feedback decoding process transforms the dataset from $D^{\text{raw}}$ to $D = \{(x^{(i)}, \hat{a}^{(i)}, p^{(i)}, \hat{\gamma}^{(i)})\}_{i=1}^{N}$. We study several learning approaches using this data: supervised learning, offline reinforcement learning (RL), or the KTO-style utility maximization (Ethayarajh et al., 2024).

**Supervised Learning** We fine-tune on positive data points ($\hat{\gamma}^{(i)} = $ `positive`) and discard data points predicted as `neutral` or `negative`. We use cross entropy loss with additional label smoothing to prevent overfitting and encourage exploration. Our setup is distinct from conventional supervised learning in that the data is coming from the model interactions (i.e., on-policy), and not from a given dataset. Also, we run the learning process iteratively, each time with more data. We do not design the supervised approach in any special way to fit these changes, but this is a potential avenue for future work, which can further improve performance.

**Reinforcement Learning** We follow prior work (Kojima et al., 2021) and use simple REINFORCE-style policy gradient (Williams, 1992). The categorical feedback $\gamma^{(i)}$ (i.e., the text generated by the prompted LLM) is mapped to a numerical value with a simple reward function:

$$R(\gamma) = \begin{cases} 1, & \gamma = \texttt{positive} \\ 0, & \gamma = \texttt{neutral} \\ -0.1, & \gamma = \texttt{negative} \end{cases}. \tag{1}$$

Dropping the $i$-superscripts for simplicity, the gradient estimator for a single example is:

$$\Delta = cR(\hat{\gamma})\nabla \log P(\hat{a}|x; \theta_{\rho+1}) \qquad c = \begin{cases} 1, & \text{if } R(\hat{\gamma}) \geq 0 \\ \frac{P(\hat{a}|x; \theta_{\rho+1})}{p}, & \text{if } R(\hat{\gamma}) < 0 \end{cases}, \tag{2}$$

where the coefficient $c$ downweights examples with negative reward by their inverse propensity score (Kojima et al., 2021). This is critical because $\lim_{P(\cdot)\to 0} \log P(\cdot) = -\infty$. In practice, we also discard data points with predicted neutral feedback ($R(\hat{\gamma}) = 0$).

We choose REINFORCE for its simplicity. The positive case reduces to be mathematically equivalent to the gradient of supervised fine-tuning (SFT), whose optimization is relatively well understood. As opposed to other methods, such as PPO (Schulman et al., 2017), REINFORCE does not require a reward model and has relatively few hyperparameters. This is critical with human-in-the-loop experiments, where broad parameter sweeps are not possible. Recent work (Ahmadian et al., 2024) also suggests REINFORCE can produce on-par results in LLMs with PPO despite its simplicity.

**Utility Maximization**    To experiment with utility maximization, we use Kahneman-Tversky Optimization (Ethayarajh et al., 2024). KTO was developed to learn from per-example binary human feedback, a scenario that fits ours well. We consider examples with decoded `positive` feedback as *desired* utterances, those with decoded `negative` feedback as *undesired*, and discard those with `neutral` feedback. We refer readers to Ethayarajh et al. (2024) for the definition of the objective.

## 5    EXPERIMENTAL SETUP

**Interaction Instantiation**    We use the KILOGRAM (Ji et al., 2022) tangram images, following Gul & Artzi (2024). KILOGRAM contains 1,013 images. We randomly split them into a main split (912 tangrams) and a development split (101 tangrams). We create interaction contexts by randomly sampling 10 tangrams, and randomly select 3–5 as targets. The development split is exclusively used for seeding the initial listener policy $\pi_{\theta_0}$, and all human-bot interactions are conducted on images from the main split, i.e., tangrams that the seed policy $\pi_{\theta_0}$ has never seen before.

**Model and Initialization**    We use IDEFICS2-8B (Laurençon et al., 2024) as our model for both the policy and feedback decoder. We fine-tune with LoRA (Hu et al., 2022). We seed the initial policy $\pi_{\theta_0}$ by fine-tuning the pretrained IDEFICS2-8B weights on a small supervised dataset of 90 successful turns from 25 human-human games constructed with the development split tangrams, augmented with 12 synthetically generated deselection turns, because while necessary for human-model interactions, deselections are rare in human-human interactions (Appendix B.2). $D_0$ is reused in continual training via rehearsal. We validate our design online with 30 main-split human-bot pilot interactions, or offline with a validation set of 344 successful main-split human-human turns (Appendix A). We use the original IDEFICS2-8B for feedback decoding, because the narrow focus of our data is likely to inhibit some general linguistic knowledge. This means we cannot see improvement in the model feedback decoding capability, likely low-balling the potential of the approach. It remains an important direction for future work to keep the decoder model in sync with the policy. This requires deployments that include high domain diversity. We observe the original IDEFICS2-8B to provide robust feedback decoding out of the box, confirming our hypothesis, and providing a solid ground for our experiments.

**System Variants**    We study six system variants based on two dimensions: (a) feedback decoder configuration (binary vs. ternary); (b) optimization methods (supervised vs. REINFORCE vs. KTO):

- B-SUP and T-SUP binary (B) / ternary (T) that only trains on positive data points with a supervised fine-tuning objective (SUP).
- B-RL and T-RL trains on both positive and negative data points using REINFORCE.
- B-KTO and T-KTO are like B-RL and T-RL, but using KTO.

For variants involving negative data points (B-RL, T-RL, B-KTO, and T-KTO), we subsample negative ones to keep the positive:negative ratio close to 5:4  (Ethayarajh et al., 2024).

**Deployment**    We conduct three rounds of training-deployment for all six systems and three more rounds for B-SUP. We select B-SUP for another three rounds because it is the most promising variant after three rounds, and we want to observe its progress over a longer period. The reason for this cascaded design is the high cost of experiments. We do not distinguish between training and evaluation in the traditional sense. Instead, all listener policies are evaluated live on MTurk on about 330 human-bot interactions each round containing roughly 2400 turns. Then the same data is used to train the next iteration of policies respectively. The policies in the same round are deployed concurrently in a randomized experiment on the same set of games to mitigate human biases and variances due to game difficulty. More details on crowdsourcing are in Appendix A.3.

**Learning Implementation Details** We use the validation set for model selection throughout continual learning. Following prior work (Misra et al., 2017; Müller et al., 2019; Liu et al., 2022), we add an entropy term and length normalization to all three objectives to reduce over-fitting given the relatively small amount of data. Appendix B provides additional reproducibility details. Unlike with REINFORCE, where we train from scratch each round, when using KTO, we continually fine-tune from a previous model checkpoint $\theta_\rho$ to obtain $\theta_{\rho+1}$ with data accumulation. This was shown to outperform training from scratch in pilot studies.

**Evaluation** We evaluate each system variant at each round by the success rate during the live deployment. We report both interaction- and utterance-level success rates. The interaction level success rate is straightforward - whether the game ended with all targets selected by the listener and nothing else. The utterance level success rate is more nuanced because we do not have access to the ground truth, i.e., the intended action. We sample 1,000 utterances per round from B-SUP to annotate by MTurk workers post hoc. We report two measures: exact match between the annotation and model action and similarity score, which is based on the computed similarity between the tangrams selected or deselected during the turn by the human annotator and the system. We also evaluate the quality of the feedback decoder by comparing its predictions with human interpretations collected during the post-hoc annotation. Because of cost, we cannot do post-hoc annotation for all system variants, so we also report click accuracy, which approximates utterance-level performance well. It measures the ratio of the model actions that lead to selection statuses that do not violate the set of targets (i.e., selections of target tangrams are good, deselection of non-target tangrams are good). Lastly, we track the number of turns per interaction. Appendix B.4 provides full definitions of our metrics.

## 6 RESULTS AND ANALYSIS

We deploy our models for three rounds, with additional three rounds for B-SUP, the best-performing variant, to better understand long-term dynamics. All our results are from concurrent randomized deployment, where the models interact with humans in real time. We collected a total of 7,230 interactions consisting of 55,004 utterances over 4 weeks, at a cost of $11,180 USD.

Figure 4 shows the deployment statistics for all six system variants, as well as control deployments for the initial policy and human-human games.[5] Figure 5 shows utterance-level statistics for B-SUP from the post-hoc annotations we collected. The interaction success rate of all systems improves monotonically in the first three rounds, except for B-KTO in round three. We conduct three more rounds with B-SUP, the leading system after the first three rounds. B-SUP then plateaus, and even shows a temporary decrease in performance, before resuming its improvement.[6]

Overall, B-SUP improves interaction-level success rate by 51% (31%→82%) and utterance-level exact match by 22% (31%→53%). At the last round, following the plateau, B-SUP interaction success rate improves by 5% (77%→82%). The number of turns follows these trends. As the policy gets better, more games are completed within the allotted number of turns, and even faster. B-SUP starts with 8.9 turns per game, and concludes with 6.7 per game. The center panel of Figure 5 shows that actions taken by the policy increasingly resemble human actions, even mistakes (actions that receive negative feedback) become more similar to human actions. All other statistics largely track these, except some of the utterance-level statistics around when B-SUP plateaus. While all show a deviation from the monotonous earlier trend, some show a temporary decrease and not just a stagnation, but delayed by one round. This illustrates the complex dynamics of continual learning, which we explore in more detail below.

There remains a significant gap between B-SUP (our leading system) and HH (human-human interactions), which shows perfect task success rate and almost double efficiency (i.e., tasks are completed with much shorter interactions). Our intuition is that the gap is due to the lack of long-term credit

---

[5]We present results in rounds for simplicity. Appendix C connects rounds to cumulative number of interactions. Appendix E presents full tables corresponding to these plots.

[6]The reasons behind the plateau are hard to infer. One hypothesis we considered is that changes in the amount of data over time made some settings sub-optimal. Specifically, we considered our LoRA adapter settings, as they impact the expressiveness of fine-tuning. We conducted a separated deployment, branching out from round three for two rounds (four and five) with B-SUP and more expressive adapters. We observed this increase in expressivity allows the model to continue its monotonous improvement. Appendix D provides the details. This mini experiment illustrates the complexities of continual learning with current learning systems.

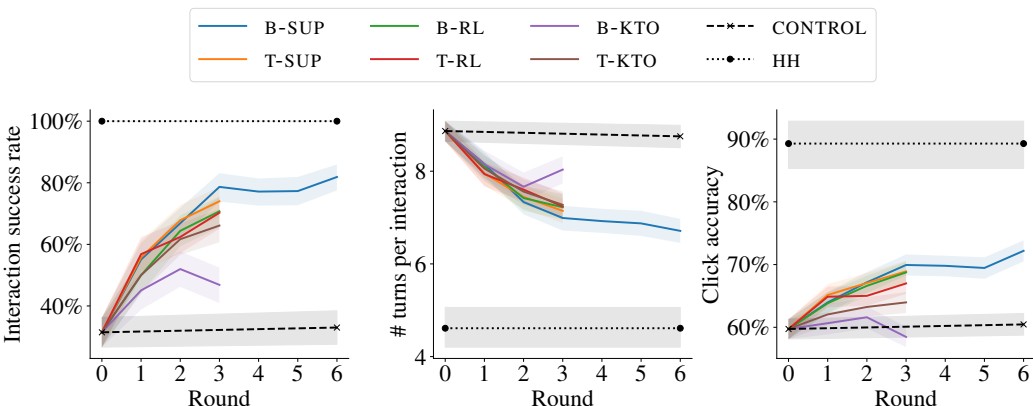

Figure 4: Task performance and efficiency improve as the policy learns from more past interactions. We present deployment results across three rounds for six concurrent systems, and three more rounds for the top system B-SUP, together with human-human references (HH) and a redeployment of the initial policy $\pi_{\theta_0}$ (CONTROL). *Left:* interaction-level success rate ($\uparrow$, higher is better). *Center:* interaction-level efficiency by # turns per interactions ($\downarrow$). *Right:* micro-level performance by click accuracy ($\uparrow$). Shades are 95% confidence intervals by bootstrapping with 10,000 resamples.

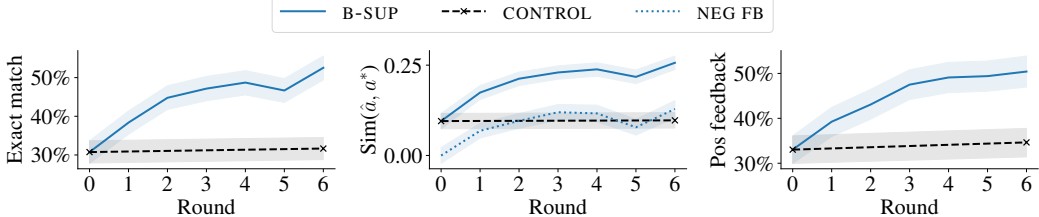

Figure 5: Turn-level performance of B-SUP evaluated by post-hoc human annotations. *Left: %* turns where the policy's action $\hat{a}$ matches exactly the human listener's action $a^*$ ($\uparrow$). *Center:* similarity between the policy's action and the human listener's action ($\uparrow$). Even actions that receive negative feedback in deployment (NEG FB) are increasingly similar to human actions. *Right: %* turns that annotated to have received positive implicit feedback from human listeners ($\uparrow$).

assignment in our learning method. This is especially influential in learning to reason about later turns. Later turns show much stronger dependence on earlier turns, creating a more complex reasoning problem and a harder credit assignment problem. This learning challenge is compounded by data scarcity: we have significantly less data for later turns, as not all interactions include them. This can potentially be addressed by not including all past turns in the context (i.e., sliding window approach).

**User Adaptation**    A potential confounder is user adaptation: the improvement in interaction success rate could have been attributed to users adapting to the interaction scenario and the system, instead of policy improvement (Hawkins et al., 2020a). We redeploy the initial policy $\pi_{\theta_0}$ concurrently to final B-SUP round to test this (CONTROL in Figure 4). The interaction success rate of CONTROL remains relatively stable over time ($31\% \rightarrow 33\%$), suggesting that speaker familiarity and adaptation do not explain the overall 51% absolute improvement in B-SUP interaction success rate.

**Positive Only vs. All Data**    The difference between systems using positive learning signals only (B-SUP, T-SUP) and those using all (B-RL, T-RL, B-KTO, T-KTO) is in learning objectives (supervised vs. RL/KTO). Overall, the systems based on positive signals only perform better. It is expected that positive signals will be more informative for learning. Our policy acts in a large action space. Negative rewards suppress specific actions, but without more information about what a good action is, they simply encourage a uniform distribution. This has been shown to have a helpful regularizing effect in past work (Kojima et al., 2021). However, not only does negative feedback not help meaningfully, it seems to confuse the learner. The positive-only systems that, in effect, have access to fewer learning signals perform better. Utilizing negative signals better is an important direction for future work.

**Binary (top row):**

|  | Pred neg | neu | pos | neg | neu | pos | neg | neu | pos | neg | neu | pos | neg | neu | pos | neg | neu | pos | neg | neu | pos |
|---|---|---|---|---|---|---|---|---|---|---|---|---|---|---|---|---|---|---|---|---|---|
| Act neg | 0.46 | - | 0.02 | 0.46 | - | 0.01 | 0.44 | - | 0.02 | 0.43 | - | 0.01 | 0.40 | - | 0.01 | 0.41 | - | 0.01 | 0.38 | - | 0.01 |
| neu | 0.12 | - | 0.07 | 0.08 | - | 0.06 | 0.06 | - | 0.05 | 0.04 | - | 0.05 | 0.05 | - | 0.05 | 0.05 | - | 0.03 | 0.06 | - | 0.04 |
| pos | 0.16 | - | 0.17 | 0.15 | - | 0.24 | 0.14 | - | 0.29 | 0.15 | - | 0.32 | 0.15 | - | 0.34 | 0.17 | - | 0.33 | 0.16 | - | 0.34 |

**Ternary (bottom row):**

| 0.38 | 0.09 | 0.01 | 0.42 | 0.04 | 0.01 | 0.41 | 0.04 | 0.01 | 0.39 | 0.04 | 0.00 | 0.33 | 0.07 | 0.01 | 0.35 | 0.05 | 0.01 | 0.34 | 0.05 | 0.01 |
|---|---|---|---|---|---|---|---|---|---|---|---|---|---|---|---|---|---|---|---|---|
| 0.07 | 0.08 | 0.04 | 0.05 | 0.05 | 0.04 | 0.04 | 0.03 | 0.04 | 0.03 | 0.03 | 0.03 | 0.03 | 0.04 | 0.04 | 0.03 | 0.03 | 0.03 | 0.03 | 0.04 | 0.03 |
| 0.07 | 0.12 | 0.14 | 0.07 | 0.12 | 0.20 | 0.06 | 0.13 | 0.24 | 0.06 | 0.15 | 0.27 | 0.04 | 0.17 | 0.28 | 0.03 | 0.20 | 0.26 | 0.04 | 0.19 | 0.27 |

$\rho = 0$    $\rho = 1$    $\rho = 2$    $\rho = 3$    $\rho = 4$    $\rho = 5$    $\rho = 6$

Figure 6: Confusion matrices of the binary (top row) and ternary (bottom row) feedback decoders over rounds. Feedback decoders yield negligibly low false positives (top right corner), even in early rounds. The feedback decoder also correctly classifies more than 60% (diagonals) across rounds.

**Feedback Decoder Evaluation**  We evaluate the quality of the feedback decoder through our annotation task. For each turn, workers annotate if the speaker was satisfied with the answer given their followup utterances. Figure 6 shows feedback decoding confusion matrices over time. The feedback decoder performance is relatively stable throughout the rounds, showing robustness to changes in the data distribution. If we collapse together actual positives and neutrals, we observe above 90% precision consistently. The ternary feedback decoder is more conservative compared to the binary one and labels more positive turns as neutrals. This is a task-dependent trade-off. The zero feedback of neutrals essentially eliminates the examples, but allows for slightly cleaner data. Here we empirically observe it is beneficial to have slightly noisy data but more of it.

**Supervised vs. REINFORCE vs. KTO**  Overall, the supervised variants (B-SUP and T-SUP) perform best. The KTO variants (B-KTO and T-KTO) trail after the REINFORCE variants (B-RL, T-RL). B-KTO even diverges at some point and starts losing performance fast. We suspect this is because the KTO recipe does not work well in the challenging optimization scenario of continual learning, where the model is fine-tuned multiple times. We observe that B-KTO deteriorates in rounds two and three, and starts generating illegal outputs (e.g., Deselect select). Appendix B.3 describes a quick intervention we applied to try to mitigate this issue. Although it eliminated the illegal outputs, the quality remained low. It is possible that further refinement of how KTO is used or further tuning of its hyperparamters will help. However, this is a complex process in a live deployment.

**Language Analysis**  We analyze the human instructions and how they change as the policy learns from more interactions (Figure 7). We observe a reduction in vocabulary size and utterance length early on. This is expected, and follows known observations in how humans adapt to reduce cognitive costs (e.g., Clark & Wilkes-Gibbs, 1986; Effenberger et al., 2021). However, in later rounds, B-SUP witnesses an increase in vocabulary size and utterance length. This surprising trend reversal is attributed to only three outlier workers, so does not express a significant change in population behavior. The number of reset signals drops, another reflection of improved collaborated task performance. Such trends are fairly consistent across system variants, except for B-KTO, which also shows divergence in performance. We observe that initially workers tend to use *Try again* instead of directly describing a target, or request a reset with instructions like *Deselect everything* (Figure 15 and Figure 16). The occurrences of both decrease in later rounds. Even though the workers change their language, this does not really help the initial policy $\pi_{\theta_0}$, which remains poor (Figure 4).

# 7 RELATED WORK

**Learning from Feedback**  Learning from feedback for LLMs is being studied extensively. RL from human feedback (RLHF) is maybe the most common technique (Ouyang et al., 2022). It relies on soliciting pair-wise preferences from annotators, which is significantly different than our reliance on unpaired signals from the interaction itself. Learning from feedback on a single system output has also been studied, either in the form of binary feedback (Ethayarajh et al., 2024; Suhr & Artzi, 2023; Gao et al., 2023) or through more expressive editing (Gao et al., 2024) or commenting and

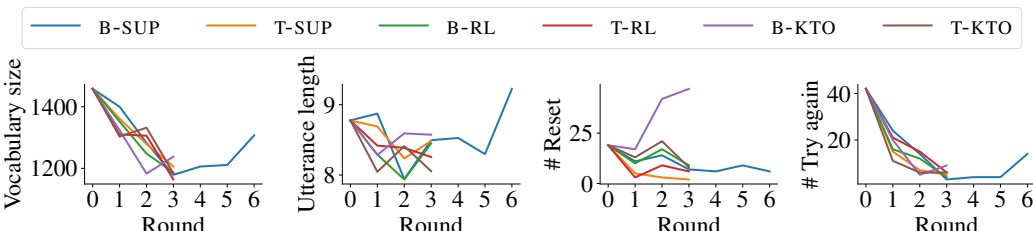

Figure 7: Language analysis of human instructions. All systems show a decrease in instruction complexity in the first three rounds, except for B-KTO, suggesting adaptation and improved efficiency on the speaker's side. Keyword-based analysis reveals that the number of reset/frustration signals drops, a reflection of the model learning and collaboration improving.

refinement (Li et al., 2017; Sumers et al., 2021; Scheurer et al., 2023). Hancock et al. (2019) trains a separate supervised model to continually predict satisfaction levels, which is then used to pause interactions and solicit explicit feedback. We do not solicit feedback, but rely on natural signals that arise from the followup interaction. Some of these include explicit feedback, but many do not.

**Learning from Naturally Occurring Signals**   Kojima et al. (2021) presents an approach to learn to generate instructions by observing how humans follow them, a complementary mode of learning to our focus on general response. Pang et al. (2023) maximizes heuristics, such as the chance of long responses from humans, in a chatbot scenario. Artzi & Zettlemoyer (2011) studied the use of naturally occurring recovery efforts (i.e., when the user switches to simpler language to relay information) to train a symbolic semantic parser from a corpus of dialogue interactions. In contrast, we opt for a general approach to infer feedback from natural language interactions of the model itself.

Concurrently to our work, Don-Yehiya et al. (2024), as well as Petrak et al. (2023), proposes an approach that uses naturally occurring feedback in conversations to filter a large conversational corpus. The linguistic cues they rely on are similar to ours. Unlike our study, the model they improve is not the model that generated the interactions, creating a distillation-like setup, where improvement is not coming from the model's own interaction, but from other models. We focus on model self-improvement, where it is critical that no stronger model is involved. Another difference is our interest in continual deployment with humans, whereas they follow a standard train-test benchmarking recipe. This allows our work to expose dynamics that are otherwise hidden. Our work and Don-Yehiya et al. (2024) complement each other and strengthen our conclusions. Their work shows the signal can be derived from large-scale diverse data, whereas ours shows how a single-model loop can work over a long period of time, and the dynamics it creates.

**LLMs that Self-improve**   A common approach to improve models is via AI feedback, solicited from the model itself or another model (Bai et al., 2022; Burns et al., 2023; Madaan et al., 2023; Kumar et al., 2024; Qu et al., 2024; Yuan et al., 2024; Li et al., 2024). In contrast, we elicit *real human* feedback automatically from the interactions in deployment. This signal is more on-the-job, and less influenced by model biases. We also use the same model for interaction and inferring feedback, ruling out concerns about distillation.

## 8   DISCUSSION

We introduce RESPECT: retrospective learning from interactions, an annotation-free approach by leveraging signals from naturally occurring feedback in interactions. We demonstrate its effectiveness in long-term deployments and robustness to system variants. As opposed to evaluating on a static benchmark, we design MULTIREF to study real interactions over a period of time. We make trade-offs between the generality of the task, and the ability to iterate on a prototype fast, and without high costs. It is important to expand this type of study to other tasks, such as summarization or conversational question answering, where similar signals may be more complex, far apart, or demand long-term credit assignment. Another interesting orthogonal direction is expanding the expressivity of the feedback decoder, such that it recovers a more expressive signal (e.g., a natural language explanation).

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

## A  THE MULTIREF GAME DESIGN AND DATA COLLECTION

### A.1  INTERACTION DESIGN

MULTIREF is a multi-target, multi-turn reference game between two players, a *speaker* and a *listener*. Each game starts with 10 tangrams as the *context*, with 3–5 tangrams designated as *targets*. The target designations are revealed to the speaker but hidden to the listener. The goal is to select all targets without selecting any non-targets. The speaker can only communicate with the listener through a sequence of utterances, and only the listener can take selection and deselection actions. The interaction starts with a speaker turn. Turns alternate between speaker and listener, with a maximum of 20 turns. In each speaker turn, they type an utterance to send to the listener. Speaker turns are limited to 25 seconds. In each listener turn, they have 45 seconds to select or deselect images as instructed to by the speaker. The game concludes when the listener selects only and all targets, or the when the partners run out of turns. Appendix A.3 shows screenshots of the interface.

**Context Construction**  We follow Gul & Artzi (2024) and construct game contexts using 1,013 tangram images from KILOGRAM Ji et al. (2022). We group tangrams randomly into two splits: development split (101 tangrams) and main split (912 tangrams). The development split is exclusively used for seeding the initial listener policy $\pi_0$. All human-bot interactions are constructed from the main split, i.e., tangrams that the seed policy $\pi_0$ has never seen before. We construct all games with 3–5 target tangrams. More targets are generally harder, given the same maximum number of turns per interaction.

### A.2  HUMAN EVALUATION DESIGN

Automatically evaluating turn-level policy performance is hard, because we have no ground truth (i.e., the selection and deselection actions intended by the speaker in each turn) to compare against. Similarly, we have no ground truth to systematically assess the feedback decoder quality. We conduct human evaluation surveys to address these problems. We annotate a subset of B-SUP interactions, roughly 120 interactions or 1,000 turns per system-turn.

We show human annotators a complete interaction turn by turn, without revealing the underlying targets. For each turn, the annotation consists for two phases:

1. Ground-truth: we show context, currently selected tangrams, and instruction given by the speaker. We ask the annotator to annotate the listener action. The annotator action $a^*$ is considered as ground truth action for this turn. We use these labels for tune-level evaluation. After the action annotation, we reveal the action $\hat{a}$ actually taken by the listener (i.e., the model) during the interaction.

2. Satisfaction: we present the follow-up utterance. We ask the annotator to rate if the speaker is satisfied with the listener's action, based on the follow-up utterance. They choose one of the following options:
   a. Yes.
   b. Yes, even though the listener did not perform all required selections/deselections.
   c. Yes, even though the listener made incorrect selections/deselections.
   d. No.

   The third option accounts for the listener accidentally selecting a target tangram not intended by the speaker, but the speaker choosing to move on without correction or even validating the selection. We treat these labels as ground truth for evaluating feedback decoders.

We annotate 5% of long-term human-bot interactions annotations by three different annotators, to estimate how reliable the annotations are. We observe 85% agreement on the correctness (whether $\hat{a} = a^*$) on ground truth stage,[7] and 65% agreement on the ground-truth action $a^*$ across workers.[8] For satisfaction annotation, we observe 93% agreement rate, illustrating the relative simplicity of extracting the signal that drive our learning process.

---

[7]The percentage of cases where all annotators agree that the bot did right or wrong.
[8]The percentage of cases where all three annotators provided exactly the same set of actions.

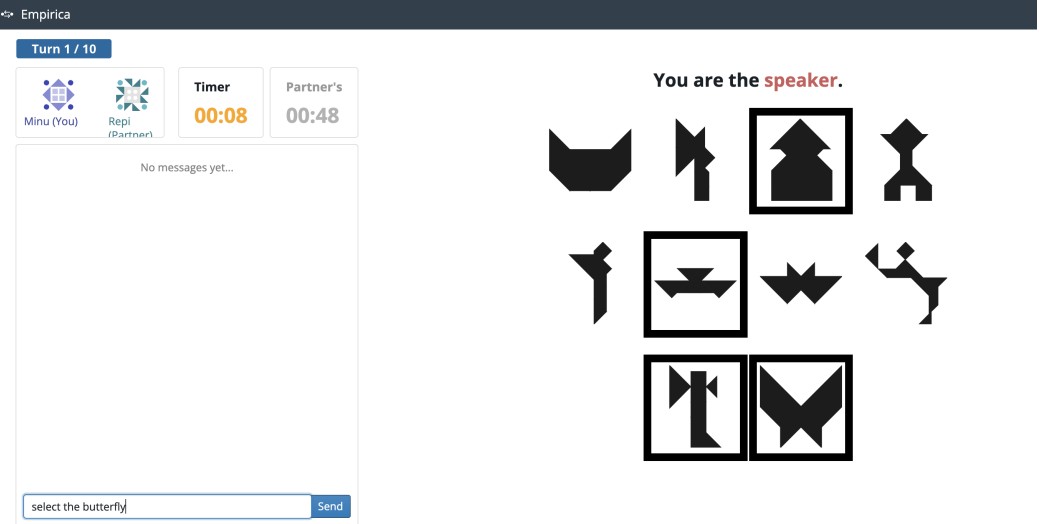

Figure 8: The MULTIREF interface for the speaker in turn 1. Predefined targets are revealed to the speaker in black boxes.

## A.3 MTURK DETAILS

**Worker Recruitment** We follow Gul & Artzi's (2024) worker recruitment recipe. We require workers to have a minimum 98% approval rate, at least 1,000 approved HITs (Human Intelligence Task), and be located in English-majority locales. All workers must watch a video tutorial and pass a quiz before gaining qualification to work on MULTIREF interactions. They must read a thorough guideline and pass another quiz before granted access to human evaluation surveys. We recruit 33 expert workers to interact with LLMs in the main study and annotate by completing surveys after the main study. This study is exempted by Institutional Review Board.

**Payment** We pay workers $0.81 USD per MULTIREF game, and a bonus if the game is successful. Overall the estimated hourly wage is $13.00 USD, and closer to $23.00 USD by the end of the continual study when the LLM is fairly good at the game. On average a human-bot game takes under 2 minutes. We pay workers $0.06 USD per turn for human evaluation surveys, or $0.08 USD if the turn annotation involves error modes. The estimated hourly wage is $16.00 USD for human evaluation surveys. On average it takes under 2.5 minutes to annotate one game. We set the payment scheme through pilot studies and aim at $15.00 USD hourly wage.

**Interface and Serving** We implement MULTIREF using Empirica (Almaatouq et al., 2021) and on top of the code base of Gul & Artzi (2024). The speaker has 25 seconds to type into a chat box each turn and hit Enter or submit, and the listener has 45 seconds to click on the tangrams to select or to deselect. The game ends if one party idles for one turn, and the party idling is not compensated. We serve on an EC2 instance. We serve LLM policies with the Ray framework (Moritz et al., 2018). We walk through the first turns of a sample interaction in Figure 8, Figure 9, and Figure 10.

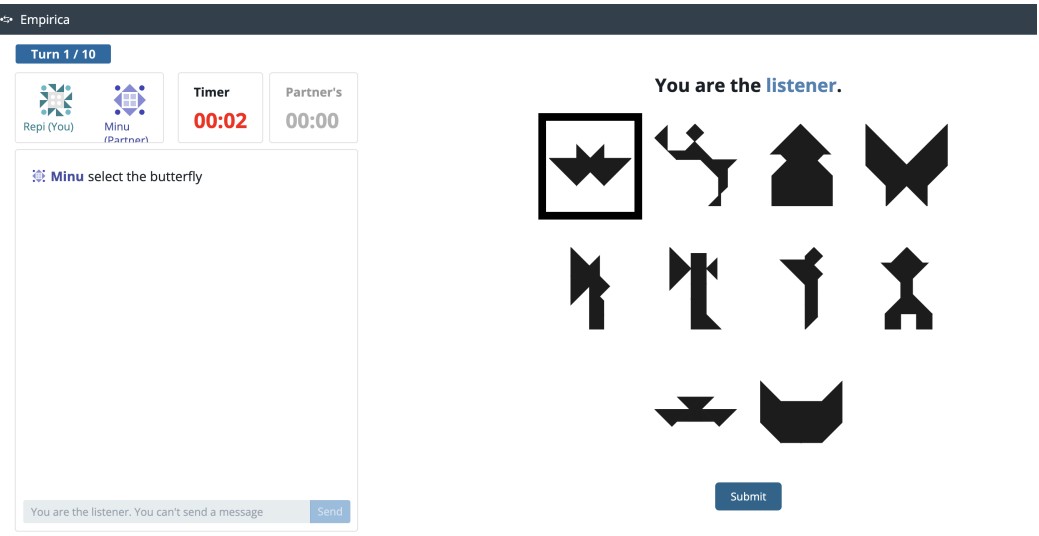

Figure 9: The MULTIREF interface for the listener in turn 2, following the speaker turn in Figure 8. Targets are hidden for the listener, and the context tangrams are in a different order. Here the listener has selected a tangram given the instruction *select the butterfly*.

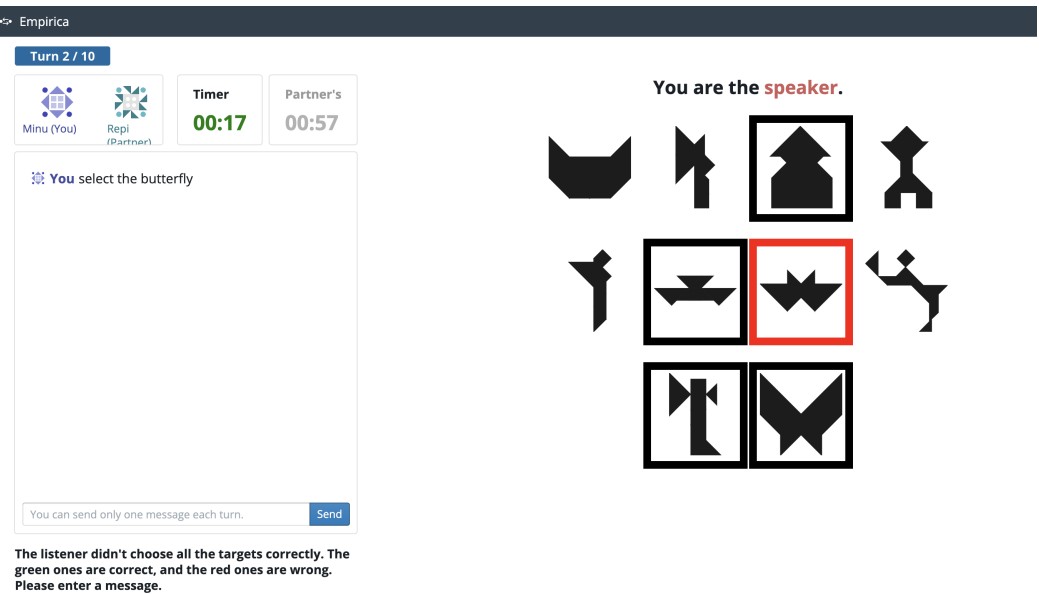

Figure 10: The MULTIREF interface for the speaker in turn 3, following the listener turn in Figure 9 The listener selected a non-target tangram, shown in red to the speaker.

---

**Policy prompt**

System: <img>A <img>B <img>C <img>D <img>E <img>F <img>G <img>H
<img>I <img>J .................... *(Context tangrams tagged by code names)*
System: none is selected ...................................... *(Turn 1 state)*
User: Select the letter S with the large triangle back side ..... *(Turn 1 utterance)*
Assistant: Select J ........................................... *(Turn 1 action)*
System: J currently selected
User: unselect that one and try again
Assistant: Deselect J select A
System: A currently selected
User: almost, unselect that one and try one more time
Assistant: Deselect A select G
System: G currently selected
User: correct. Now select the rabbit facing right
Assistant: Select F
System: F G currently selected
User: no that is a bird, try again
Assistant: Deselect F select H
System: G H currently selected
User: no that is another bird, try again
Assistant: **Deselect H select F** ....................................... *(Action)*

Figure 11: Policy prompt example with a model predicted **action** and additional *comments* for readability.

## B    LEARNING DETAILS

### B.1    INTERACTION REPRESENTATION

We encode the context $x$ as in Figure 11. We standardize action representation by ordering actions, for example, always produce `Select A C` rather than `Select C A`. We shuffle the context images during training as the order of context tangrams should not have any impact on the interaction logic.

### B.2    POLICY INITIALIZATION

We seed the initial policy $\pi_0$ by fine-tuning the model on a small dataset of 90 turns $D_0$, where both the speaker and the listener are humans. We also experimented with prompting to initialize the policy. We find early that few-shot prompting yields a random policy at best, likely because reasoning with abstract shapes such as tangrams is visually out-of-distribution for the model.

There is a significant distribution shift between human-human interactions, and human-policy interaction, especially early on when the model performs poorly. In practice, two major differences are the length of interactions and the prevalence of deselection instructions, which are rare in human-human interactions. We address the deselection issue with data augmentation. We synthetically generate turns where the speaker asks for deselections, and the listener complies. We augment the data with these at a ratio of 1:12 to the existing data. This helps the LLM policy learn to deselect and recover from mistakes. This augmentation is only used for $D_0$ and such distribution shift is not present in alter rounds, when learning from actual human-bot interactions.

### B.3    HYPERPARAMETERS AND OTHER IMPLEMENTATION DETAILS

We use the instruction-tuned IDEFICS2-8B model for all policies. We fine-tune with LoRA adapters (Hu et al., 2022) ($\alpha=r=8$, dropout=0.1) due to compute constraints. Appendix D provides more LoRA details. We train each model with a single GPU, RTX A6000, NVIDIA A100 40GB or 80GB. The time to train ranges between 2–24 hours, longer in later rounds as more data accumulates. For stopping criteria, we pick checkpoints by highest accuracy (exact match) among

---

**Policy prompt with deselection augmentation**

[Previous turns omitted]
System: none is selected . . . . . . . . . . . . . . . . . . . . . . . . . . . . . . . . . . . . . *(Previous turns)*
Speaker: Man in a hat
Listener: Select A
System: A currently selected . . . . . . . . . . . . . . . . . . . . . . . . . . . . . *(Augmented state)*
Speaker: Wrong, undo what you selected . . . . . . . . . . . . . . *(Augmented utterance)*
Listener: **Deselect A** . . . . . . . . . . . . . . . . . . . . . . . . . . . . . . . . . . . . *(Augmented action)*

---

Figure 12: An example of deselection augmentation with augmented **action** and *comments*.

three seeds on a hold-out validation set of 344 turns $D_{\text{val}}^{\text{HH}}$. The validation set is curated from 92 human-human games the main split of tangrams. We summarize hyperparameters in Table 1.

| Hyperparameter | Search Space | Supervised | REINFORCE | KTO |
|---|---|---|---|---|
| Optimizer | | AdamW | AdamW | RMSProp |
| Learning rate | {1e-6, 1e-5, 1e-4, 2e-4} | 1e-4 | 1e-4 | 1e-5 |
| Learning rate decay | {no, cosine, linear} | cosine | cosine | no |
| Epochs | {5, 10, 20, 40} | 20 | 20 | 20 |
| Warm-up steps | {0, 10, 50} | 10 | 10 | 10 |
| Weight decay | {0, 0.01, 0.1} | 0.01 | 0.01 | 0.01 |
| Effective batch size | {16, 32, 48, 64, 128} | 64 | 64 | 64 |
| Entropy weight | {0, 0.01, 0.5, 0.1} | 0.01 | 0.01 | 0.1 |
| $\beta_{\text{KTO}}$ | {0.01, 0.1, 0.5} | | | 0.5 |
| Temperature | | 1 | 1 | 1 |

Table 1: Hyperparameter settings.

**Data Imbalance** The decoded feedback is imbalanced, with more negative examples than positive examples (3:1 to 2:1), especially at early rounds of continual learning. We address this by weighing the loss by the absolute value of the reward, i.e., $-0.1$ for RL or $\lambda_d$ and $\lambda_u$ for KTO, and by downsampling negative examples per batch, such that the number of positive examples and negative examples is roughly 5:4.

**KTO Stability** Deviation from the original KTO implementation by higher learning rate, higher $\beta$, more epochs, produce better results empirically on the validation set in pilot and round $\rho = 1$. However, in round $\rho = 2$, B-KTO policy start to degenerate by producing nonsensical actions such as `Deselect A select A B` or `Deselect select select`. We attempt to mitigate this issue during training round $\rho = 3$ by switching from weighing $\lambda_d = 4$ and $\lambda_u = 1$ as recommended in Ethayarajh et al. (2024) to $\lambda_d = \lambda_u = 1$, plus downsampling negative examples. We also introduce regex-based constrained decoding to prevent nonsensical actions for B-KTO and T-KTO policies in round $\rho = 3$. Despite that, the KTO group performs worse in live interactions (Figure 4). We suspect KTO is more challenging to optimize for iterative continual learning, but we suspect further tuning (with higher computational costs) can reduce or even eliminate these issues.

### B.4 EVALUATION METRICS

**Interaction-level Metrics** Interaction performance and statistics are computed automatically from live deployment interactions. They do not require further annotation.

1. **Success rate** = # successful interactions / # all interactions. An interaction is successful if the listener selects all and only targets before running out of 10 turns. This is the primary metric we use to evaluate the performance of the LLM policy.

2. **# Turns per interaction**. This is a measure of collaborative efficiency.

**Turn-level Metrics with Reference to Human Annotation**    We compute turn-level metrics either with respect to HH games where we consider human listener action as ground truth (e.g., $D_{\text{val}}^{\text{HH}}$), or with respect to B-SUP games where we consider actions $a^*$ annotated in post-hoc surveys as ground truth. When computed with live interactions, these metrics are biased towards longer or failed interactions because they have more turns than successful interaction.

1. **Exact match** = # exact match / # all turns. An exact match is when the action taken by the policy matches exactly the action labeled/taken by human listeners ($\hat{a} = a^*$).

2. **Similarity** = $\text{Sim}(\hat{a}, a^*)$ is a composite metric. Let $f(p, q) : \mathcal{I} \times \mathcal{I} \to \mathbb{R}$ be a function that evaluates the similarity of between two images $p, q \in \mathcal{I}$. Let the action taken by policy be $\hat{a} = \{\hat{p}_1, \hat{p}_2, ..., \hat{p}_{\hat{n}}, \hat{q}_1, \hat{q}_2, ..., \hat{q}_{\hat{m}}\}$ where $p$ are the selected tangrams and $q$ are the deselected tangrams. Denote the ground truth actions as $a^* = \{p_1^*, p_2^*, ..., p_{n^*}^*, q_1^*, q_2^*, ..., q_{m^*}^*\}$. The similarity between two actions is defined as:

$$\text{Sim}(\hat{a}, a^*) = \frac{1}{\hat{n}n^* + \hat{m}m^*} \left( \Sigma_{i=1}^{\hat{n}} \Sigma_{j=1}^{n^*} f(\hat{p}_i, p_j^*) + \Sigma_{i=1}^{\hat{m}} \Sigma_{j=1}^{m^*} f(\hat{q}_i, q_j^*) \right) \quad .$$

   If only one of $\hat{n}$ and $n^*$ is zero, we rewrite $\Sigma_{i=1}^{\hat{n}} \Sigma_{j=1}^{n^*} f(\hat{p}_i, p_j^*)$ with $-\max(\hat{n}, n^*)$, and $\hat{n}n^*$ in the denominator with $\max(\hat{n}, n^*)$, intuitively assigning -1 for each missed selection. This edge case is similarly treated for $\hat{m}$, $m^*$ and deselection. We compute similarities using embeddings from the tangram fine-tuned CLIP model of Ji et al. (2022) .

3. **Positive feedback** = # turns receiving positive feedback / # all turns. An action receives positive feedback if speaker is satisfied with the listener's action in the followup interaction. This is labelled in human evaluation survey.

**Micro-level Metric with Reference to Ground Truth Targets**    We compute click accuracy with respect to the ground truth targets (instead of the targets intended by the speaker). This is cheaper because it does not require human annotation, so we can compute it for all system variants and all interactions. However, this measure produces false positives when an action selects a target not intended by the speaker. In practice, though, we find it correlates well with our human-annotated evaluation.

We compute click accuracy for a turn given its context $x$ and action $\hat{a}$. We denote the set of ground truth targets in this interaction as $\mathcal{T}$, the set of currently selected context tangrams as $\mathcal{S}$, then for each click $c$ in $\hat{a}$ (select or deselect), we compute the click accuracy as:

$$\text{Click accuracy}(c, \mathcal{T}, \mathcal{S}) = \begin{cases} 1 & \text{if } (c \in \mathcal{T} \wedge c \notin \mathcal{S}) \vee (c \notin \mathcal{T} \wedge c \in \mathcal{S}) \\ 0 & \text{otherwise} \end{cases}$$

Intuitively, a click is approximately accurate if it selects a target or deselects a non-target. We compute this for all clicks from all interactions in a round for all systems in Figure 4.

**Corpus-level Metrics**    We analyze speaker instructions per system-round. The keyword used to generate the analysis in Figure 7 are:

1. **# Reset** = occurrences of phrases in {*reset, restart, from scratch, all over, start over, deselect everything, deselect all, remove everything, remove all, clear everything, clear all, unselect everything, unselect all, drop everything, drop all* }

2. **# Try again** = occurrences of phrases in {*try again, try one more time, the other one* }

## C    CUMULATIVE NUMBER OF INTERACTIONS OBSERVED

The main text includes results by round. We collect roughly 330 interactions per policy per round. Due to the uncertainty of live data collection, we do not always hit this exact number for each variant and round. Figure 13 shows the cumulative number of human-bot interaction seen by a policy variant by each round.

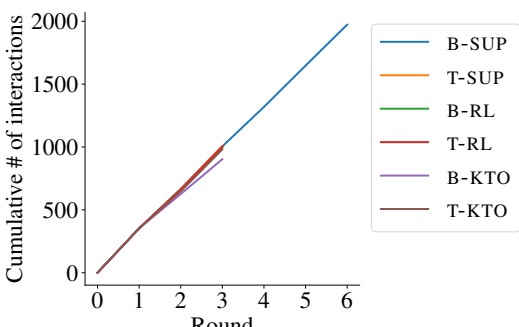

Figure 13: Cumulative number of human-bot interactions used to train the policy each round.

## D    ADDITIONAL ENHANCED LoRA LAUNCH

We suspect the plateau of B-SUP in Figure 4 is partially due to the limited expressivity of LoRA adapters we used. We test this hypothesis by deploying round $\rho = 4$ and $\rho = 5$ again with enhanced LoRA adapters. We use the same hyperparameters as in Section B.3 except additional adapters. The original adapter placement is on the text model, the modality projector, and the perceiver resampler. Adapters include the down projection layers, the gate projection layers, the up projection layers, and the key/query/value projection layers. In comparison, the enhanced launch adds adapters on the vision model, including the out projection, the first and the second fully connected layers, besides the projection layers on text models. Figure 14 shows the results from this complementary deployment. The enhanced LoRA adapters yield a small improvement in interaction success rate compared to the original launch, yet the overall slowdown is evident. This suggests LoRA expressivity has some effect, but other effects are also limiting the LLM policy from continuing its earlier improvement trends.

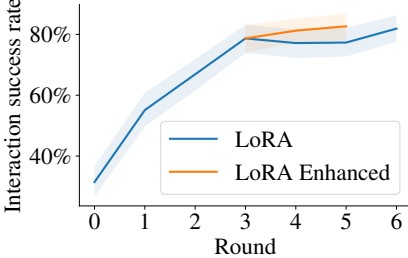

Figure 14: Success rate of B-SUP with additional LoRA adapters in round 4 and 5.

# E    DETAILED RESULTS

We present numerical results of metrics for interaction level performance in Figure 2 (Table 2, Table 3, Table 4), human evaluation performance in Figure 5 (Table 5, and language analysis in Figure 7 (Table 6, Table 7, Table 8, Table 9).

| Round | 0 | 1 | 2 | 3 | 4 | 5 | 6 |
|---|---|---|---|---|---|---|---|
| B-SUP | 31.4 | 55.1 | **66.9** | **78.7** | 77.1 | 77.3 | 81.9 |
| T-SUP | 31.4 | 55.8 | 67.8 | 74.0 | - | - | - |
| B-RL | 31.4 | 50.0 | 64.4 | 70.7 | - | - | - |
| T-RL | 31.4 | **56.8** | 62.4 | 70.3 | - | - | - |
| B-KTO | 31.4 | 45.1 | 52.0 | 46.9 | - | - | - |
| T-KTO | 31.4 | 50.0 | 61.7 | 66.1 | - | - | - |
| CONTROL | 31.4 | - | - | - | - | - | 33.0 |
| HH | - | - | - | - | - | - | 100.0 |

Table 2: Interaction task success rate in percentage (↑). We collect roughly 330 human-bot games per datapoint, except for HH where we only collect 50 games. Round 0 is shared among systems, except for HH. All system are deployed for three rounds, and the top performing one (B-SUP) is deployed for additional three rounds; preempted or not-applicable rounds are marked with dash (-). We **bold** the highest task success rate in a round.

| Round | 0 | 1 | 2 | 3 | 4 | 5 | 6 |
|---|---|---|---|---|---|---|---|
| B-SUP | 8.87 | 8.16 | **7.33** | **6.99** | 6.92 | 6.87 | 6.71 |
| T-SUP | 8.87 | 7.95 | 7.44 | 7.14 | - | - | - |
| B-RL | 8.87 | 8.10 | 7.42 | 7.22 | - | - | - |
| T-RL | 8.87 | **7.94** | 7.60 | 7.24 | - | - | - |
| B-KTO | 8.87 | 8.15 | 7.66 | 8.03 | - | - | - |
| T-KTO | 8.87 | 8.06 | 7.56 | 7.27 | - | - | - |
| CONTROL | 8.87 | - | - | - | - | - | 8.75 |
| HH | - | - | - | - | - | - | 4.61 |

Table 3: # turns per interaction (↓). Maximum 10 turns. Each game has 3-5 targets and HH games usually take one turn per target. We **bold** the fewest # turns per interaction in a round.

| Round | 0 | 1 | 2 | 3 | 4 | 5 | 6 |
|---|---|---|---|---|---|---|---|
| B-SUP | 59.7 | 64.0 | **67.2** | **69.9** | 69.8 | 69.5 | 72.2 |
| T-SUP | 59.7 | **65.2** | 67.1 | 68.9 | - | - | - |
| B-RL | 59.7 | 63.8 | 66.6 | 68.8 | - | - | - |
| T-RL | 59.7 | 64.9 | 65.0 | 67.0 | - | - | - |
| B-KTO | 59.7 | 60.7 | 61.6 | 58.5 | - | - | - |
| T-KTO | 59.7 | 62.1 | 63.2 | 64.0 | - | - | - |
| CONTROL | 59.7 | - | - | - | - | - | 60.5 |
| HH | - | - | - | - | - | - | 89.3 |

Table 4: Click accuracy in percentage (↑). We **bold** the highest click accuracy in a round.

| Round | 0 | 1 | 2 | 3 | 4 | 5 | 6 | CONTROL | HH |
|---|---|---|---|---|---|---|---|---|---|
| Exact match | 30.7 | 38.4 | 44.8 | 47.2 | 48.7 | 46.7 | 52.3 | 31.7 | 79.1 |
| Pos Feedback | 33.0 | 39.2 | 43.1 | 47.5 | 49.1 | 49.4 | 50.4 | 34.6 | 78.4 |
| $\text{Sim}(\hat{a}, a^*)$ | 19.0 | 34.8 | 42.5 | 46.0 | 47.7 | 43.5 | 51.3 | 19.4 | 83.8 |
| $\text{Sim}(\hat{a}, a^*)$ -FB | 0.0 | 13.6 | 19.2 | 23.9 | 23.3 | 15.6 | 25.7 | 1.4 | 67.9 |

Table 5: Turn level performance of B-SUP based on human evaluation, all in percentages (↑).

| Round | 0 | 1 | 2 | 3 | 4 | 5 | 6 |
|---|---|---|---|---|---|---|---|
| B-SUP | 1458 | 1400 | 1283 | 1179 | 1206 | 1211 | 1307 |
| T-SUP | 1458 | 1361 | 1279 | 1206 | - | - | - |
| B-RL | 1458 | 1352 | 1248 | 1187 | - | - | - |
| T-RL | 1458 | 1310 | 1306 | 1164 | - | - | - |
| B-KTO | 1458 | 1324 | 1183 | 1238 | - | - | - |
| T-KTO | 1458 | 1303 | 1332 | 1184 | - | - | - |
| CONTROL | 1458 | - | - | - | - | - | 1311 |
| HH | - | - | - | - | - | - | 433 |

Table 6: Vocabulary size of different systems across rounds.

| Round | 0 | 1 | 2 | 3 | 4 | 5 | 6 |
|---|---|---|---|---|---|---|---|
| B-SUP | 8.78 | 8.87 | 7.94 | 8.49 | 8.53 | 8.30 | 9.23 |
| T-SUP | 8.78 | 8.69 | 8.24 | 8.49 | - | - | - |
| B-RL | 8.78 | 8.29 | 7.94 | 8.45 | - | - | - |
| T-RL | 8.78 | 8.42 | 8.39 | 8.26 | - | - | - |
| B-KTO | 8.78 | 8.29 | 8.59 | 8.57 | - | - | - |
| T-KTO | 8.78 | 8.05 | 8.41 | 8.05 | - | - | - |
| CONTROL | 8.78 | - | - | - | - | - | 8.19 |
| HH | - | - | - | - | - | - | 8.49 |

Table 7: Utterance length of different systems across rounds.

| Round | 0 | 1 | 2 | 3 | 4 | 5 | 6 |
|---|---|---|---|---|---|---|---|
| B-SUP | 19 | 11 | 14 | 7 | 6 | 9 | 6 |
| T-SUP | 19 | 5 | 3 | 2 | - | - | - |
| B-RL | 19 | 10 | 17 | 9 | - | - | - |
| T-RL | 19 | 3 | 9 | 6 | - | - | - |
| B-KTO | 19 | 17 | 42 | 47 | - | - | - |
| T-KTO | 19 | 13 | 21 | 8 | - | - | - |

Table 8: # Reset words of different systems across rounds.

| Round | 0 | 1 | 2 | 3 | 4 | 5 | 6 |
|---|---|---|---|---|---|---|---|
| B-SUP | 42 | 24 | 14 | 3 | 4 | 4 | 14 |
| T-SUP | 42 | 15 | 7 | 5 | - | - | - |
| B-RL | 42 | 16 | 12 | 4 | - | - | - |
| T-RL | 42 | 21 | 15 | 6 | - | - | - |
| B-KTO | 42 | 20 | 5 | 9 | - | - | - |
| T-KTO | 42 | 11 | 6 | 6 | - | - | - |

Table 9: #Try again words of different systems across rounds.

## F  FEEDBACK DECODER DESIGN

The prompt design is minimal, general, and task-agnostic. We validate the prompt with manual inspection prior to continual learning launch and human surveys. Considering only the most recent two action-utterance turns $\langle \hat{a}_{i-1}, u_i, \hat{a}_i, u_{i+1} \rangle$ is sufficient to produce satisfactory decoding results, and more history seems to distract the decoder.

We also experimented with numerical reward (i.e., decoding a real number), experimenting with a discretized reward space of $\{.0, .1, .5, .9\}$ . Our experiments show the model is not well calibrated for such decoding.

## G  INTERACTION CASE STUDIES

Figures 15–18 show case studies that illustrate the diversity of MULTIREF interaction scenarios. Black borders indicate targets. Yellow dots indicate actions taken by the listener. Green borders indicate correct selections, while red borders indicate wrong selection.

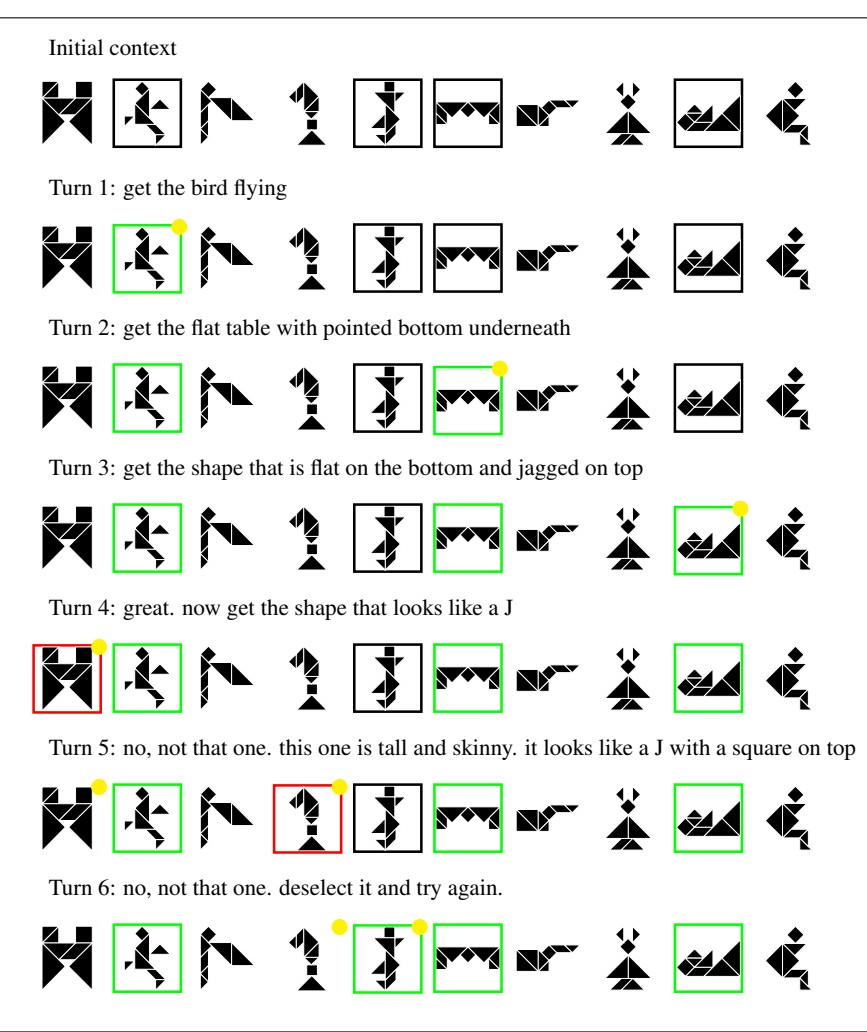

Figure 15: The speaker is left with the last target at Turn 4. Failing, they provide an additional description in Turn 5, and eventually resort to "try again" without describing the target in Turn 6. The initial turns illustrate how feedback is implied, rather than specified explicitly. The interaction concludes successfully.

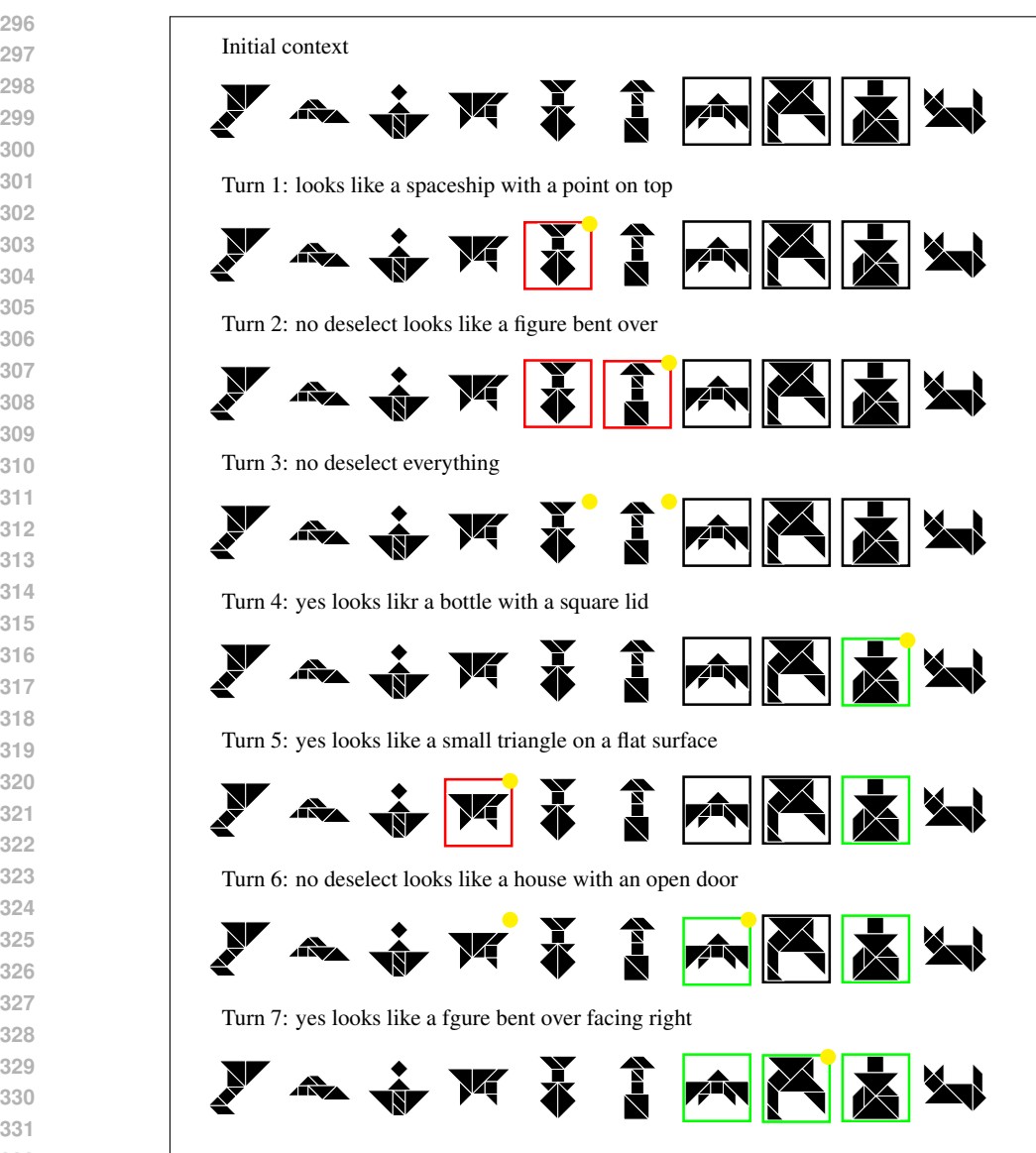

Figure 16: The speaker asks to deselect everything in Turn 3 to reset, an expression of frustration. The interaction concludes successfully.

## H    FEEDBACK DECODER ERROR AND POTENTIAL FIX

Roughly 15% of feedback decoder predictions are false negatives, see Figure 6 top row, and an example in Figure 19. We handle negatives in different ways in our experiments, but generally negatives examples have less impact than positive ones, so the learner is robust to false negative noise. Of course, it does mean that we are losing valuable positive data, and reducing this error rate is an important direction for future work. This can potentially speed up learning further.

## I    ETHICAL CONSIDERATIONS

Deploying our approach to learning from human-model interactions suffers the same risks as approaches that fine-tune on interaction data. It is critical to remove sensitive and private information by data filtering or other techniques.

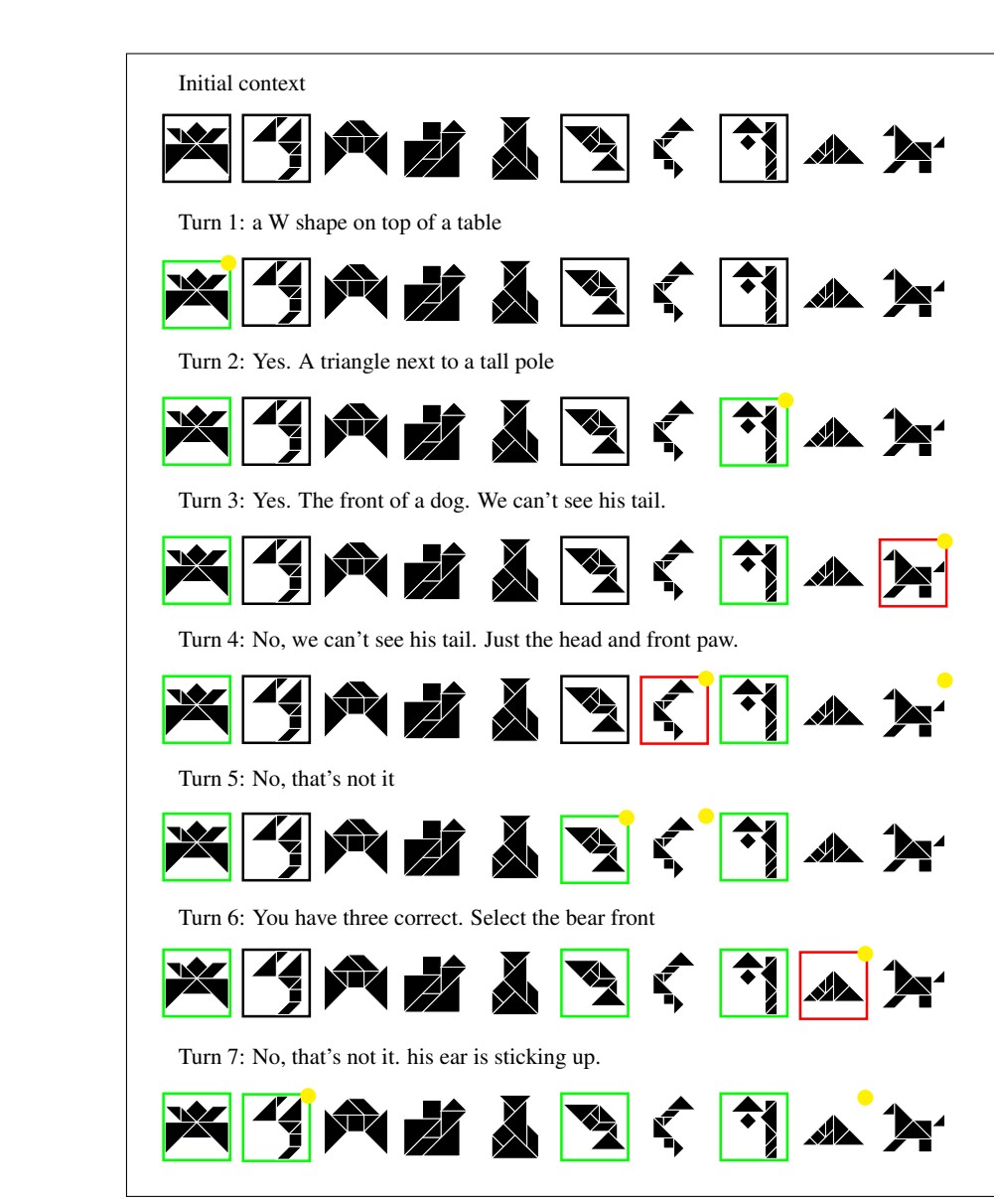

Figure 17: The abstractness and ambiguity of tangrams lend to complex interactions. There are two dogs in the context, and the listener struggles to disambiguate or identify the target. The interaction concludes successfully.

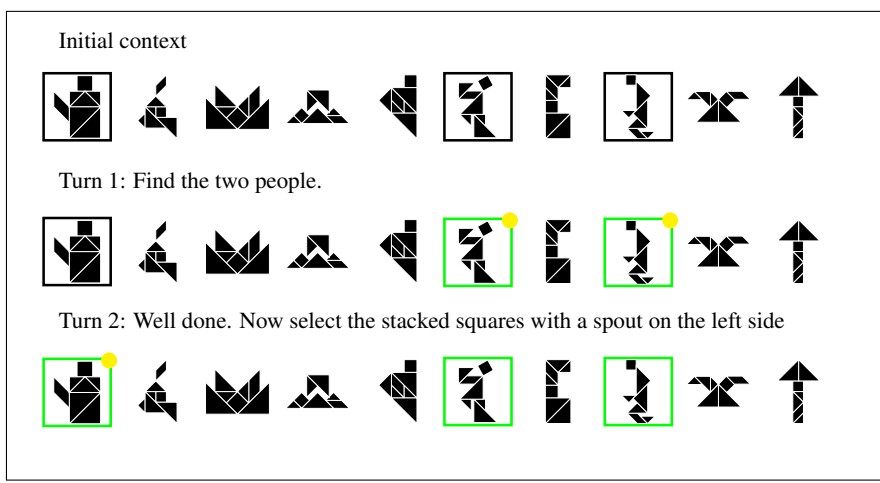

Figure 18: The speaker asks for two targets in Turn 1, exemplifying Grice's Maxims of Quantity - one tries to be as informative as one possibly can, and gives as much information as is needed, and no more (Grice, 1975). The interaction concludes successfully.

---

**Feedback Decoder Case Study (False Negative)**

User: Please carefully read the following conversation and answer: Is the very last utterance from the speaker positive or negative feedback? Often negative feedback include corrections and keywords like no, not, undo, don't, with generally negative sentiment, while positive feedback often includes good, yes, correct, okay, or simply move on to the next stage. Lean towards negative if it sounds neutral.
(start of the conversation)

       Speaker: house

       Listener: Select F . . . . . . . . . . . . . . . . . . . . . . . . . . . . . . . . . . . . . . . . . . . *(Action to focus on)*

       Speaker: horned roof . . . . . . . . . . . . . . . . . . . . . . . . . . . . . . . . . . . . . . . . . . . . . . *(Feedback)*

(end of the conversation)
Answer a single word, Positive, or Negative
Assistant: **Negative**

---

Figure 19: Feedback decoder false-negative example: the feedback decoder fails to recognize an implicit positive feedback from the speaker by moving on to the next target. The verbal **feedback generated by the model** is in bold. Additional *comments for readability* are in italics.

