# OpenReview forum: "Retrospective Learning from Interactions"
_ICLR.cc/2025/Conference — Submitted to ICLR 2025_

### Official Review · Reviewer_Aocp · 2024-11-01

**Soundness:** 3
**Presentation:** 3
**Contribution:** 4
**Rating:** 6
**Confidence:** 5

**Summary:**

In this paper, the authors propose RESPECT, a new method for refining multimodal language models (in this case vision and language models) from interaction data automatically generated by a model while interacting with another agent when solving a referential task.

To tackle this problem, the authors propose a new benchmark called MultiREF which requires agents to manipulate tangrams, specific abstract shapes that are well-known in the community for their ability to elicit interesting communicative grounding phenomena due to their intrinsic ambiguity.

Based on this dataset, the authors focus on a very specific training regime which alternates two phases: 1) retrospection = decoding implicit feedback from past interactions by means of a classifier which derives feedback labels (i.e., positive, neutral, negative); 2) learning = refining the model using the feedback received from the previous stage. Because the authors simplify the prediction task to a classification task, they argue that Step 1) can be simply performed by a carefully prompted model to perform a simple binary/three-way classification task. For the second step, the authors test different learning strategies such as a) supervised learning from positive data only; b) Online reinforcement learning (using REINFORCE) with a hand-crafted reward function which leverages the labels derived by the classifier from Step 1. c) Kahneman-Tversky Optimisation (KTO) as a form of reinforcement learning from feedback (in this case, AI feedback).

The authors set up a really complex evaluation with real users that interact with the system in real-time. In their evaluation, they start from IDEFICS2-8B model as their initialisation and use a frozen IDEFICS2-8B as the feedback decoder. From their evaluation, seems that there is still a long way to go to develop robust training regimes that can facilitate the type of adaptation required for these interactive tasks. In fact, the supervised learning variant seems to be the most robust which relies purely on positive examples and ignores negative ones.

**Strengths:**

1. Interesting evaluation that tests the system with real users over a period of 4 weeks. This represents a great effort to showcase the strengths and weaknesses of the different training regimes.
2. Very interesting idea to simplify the task of the "critic" to fixed labels that can be used for very specific training regimes
3. The authors test different training regimes that are well known in the community such as Supervised Learning, REINFORCE and KTO

**Weaknesses:**

1. Although I appreciate the rationale behind using tangrams, I wonder whether the authors could have tested this approach in more realistic reference game that are well-known in the community such as 20Q game [1], GuessWhat?! [2] or Photobook [3]. I feel like this would have given a much broader perspective on the robustness and reliability of the proposed training regimes for more complex language generation tasks

2. It's not clear to me what is the rationale behind using KTO compared to DPO which is more established (e.g., used by Meta for Llama 3.2 tuning). Considering that the authors have access to positive and negative examples, I wonder whether they should try DPO instead considering that has been tested more for VLMs

3. It's not clear to me how the authors complete their fine-tuning considering that most of the training regimes are not designed for dialogue data specifically. See Question 1 as well for details.

4. I think the authors are missing a simpler baseline that is fine-tuned using the final reward (i.e., whether you win or not the game) as a reward as was done in previous work [4].

5. The related work cites some interesting work related to using AI-generated feedback for improvement. However, the authors do not provide a baseline where this is explored for this game. For instance, the method proposed by Yuan et al 2024.


## Additional suggestions to improve the manuscript

I would suggest the author refine their manuscript by clarifying some aspects of their methodology by answering my questions below. At the same time, I would suggest them to implement the following changes:

- Extend the analysis of the paper with average dialogue turns over the different rounds; this can shed light on the ability of the models to improve their gameplay strategy over time

- Clarify how you pair different users on MTurk to maximise usage. For example, there are dedicated methods to do so (e.g., [5]) but it's not clear to me how you've arranged this.

- Provide more details regarding the REINFORCE baseline

- Typo on Line 244 "process transformers"

- It's confusing to see that you call your models (which are Vision and Language Models) as LLMs

- I would suggest improving Figure 1 to show the type of feedback that the system is leveraging for the improvement.

## References

[1]: Bertolazzi, L., Mazzaccara, D., Merlo, F., & Bernardi, R. (2023, September). Chatgpt’s information seeking strategy: Insights from the 20-questions game. In Proceedings of the 16th International Natural Language Generation Conference (pp. 153-162).

[2]: De Vries, H., Strub, F., Chandar, S., Pietquin, O., Larochelle, H., & Courville, A. (2017). Guesswhat?! visual object discovery through multi-modal dialogue. In Proceedings of the IEEE Conference on Computer Vision and Pattern Recognition (pp. 5503-5512).

[3]: Haber, J., Baumgärtner, T., Takmaz, E., Gelderloos, L., Bruni, E., & Fernández, R. (2019, July). The PhotoBook Dataset: Building Common Ground through Visually-Grounded Dialogue. In Proceedings of the 57th Annual Meeting of the Association for Computational Linguistics (pp. 1895-1910).

[4]: Strub, F., De Vries, H., Mary, J., Piot, B., Courvile, A., & Pietquin, O. (2017, August). End-to-end optimization of goal-driven and visually grounded dialogue systems. In Proceedings of the 26th International Joint Conference on Artificial Intelligence (pp. 2765-2771).

[5]: Götze, J., Paetzel-Prüsmann, M., Liermann, W., Diekmann, T., & Schlangen, D. (2022, June). The slurk Interaction Server Framework: Better Data for Better Dialog Models. In Proceedings of the Thirteenth Language Resources and Evaluation Conference (pp. 4069-4078).

**Questions:**

1. It's not clear to me how the authors complete their fine-tuning considering that most of the training regimes are not designed for dialogue data specifically. For instance, if you have a dialogue of 4 turns, do you simply treat this as a single example or do you derive many examples for it? This is an important detail which I don't think is specified in the section that describes the training regime.

2. What kind of REINFORCE implementation did you use? Did you adopt a baseline term? I think it's important to report more detail to aid reproducibility

3. Considering that the action space of the model is very limited, have you considered a form of token masking to improve the performance of your algorithms?

---

> ### Author Response · Authors · 2024-11-20
> **Initial Response Part 1**
>
> We thank the reviewer for their helpful comments. We address the concerns below, and following this discussion, we will upload the updated version (the changes are not yet in the PDF).
>
> ## Response to Raised Weaknesses
>
> > Although I appreciate the rationale behind using tangrams, I wonder whether the authors could have tested this approach in more realistic reference game that are well-known in the community such as 20Q game [1], GuessWhat?! [2] or Photobook [3].
>
> MultiRef has several key attributes of language interaction that are not well exposed by the other scenarios you list (although we do consider them interesting and related). MultiRef has a combinatorial solution space. This elicits the multi-turn interaction, with gradual instruction by the speaker and gradual solution construction by the listener. The impact of listener's actions is immediately seen by the speaker. The gradual construction and immediate visibility are essential to elicit the natural implicit feedback we observe. These are properties that are common in natural interactions.
>
> 20Q and GuessWhat both are focused on binary questions, in contrast to our instructions that lead to more complex action space. They also both feature a much smaller solution space, which is guessed in one goal after the data is shared, not allowing for the immediate visibility of action impact after each turn. PhotoBook is more related to our scenario, but it also doesn’t provide real-time action visibility, and actions are only visible after both sides submit. More important: PhotoBook is a collaborative scenario where two equal partners negotiate a common ground. MultiRef is an instructional scenario, where the speaker instructs the listener.
>
> Therefore, while the scenarios you list are relevant, their focus is different, and they don’t expose the kind of real-time dynamic behavior we observe in real-time multi-turn interactions – and this is what MultiRef is about.
>
> We will add a discussion of the related scenarios.
>
> > It's not clear to me what is the rationale behind using KTO compared to DPO which is more established (e.g., used by Meta for Llama 3.2 tuning). Considering that the authors have access to positive and negative examples, I wonder whether they should try DPO instead considering that has been tested more for VLMs
>
> DPO requires paired preferences, that’s **two** potential outputs for exactly the same input, one marked as winner and the other as loser. We don’t have this kind of data, so DPO is not applicable. This is why we chose KTO.
>
> > It's not clear to me how the authors complete their fine-tuning considering that most of the training regimes are not designed for dialogue data specifically. See Question 1 as well for details.
>
> We create an example for each utterance. Each such example includes the set of images, their selection status,  the history of the interaction up to the current utterance, and the utterance. The model output is the action generated by the model. The learning algorithm sees each such “example” as separate. This data formulation is identical to what the model “sees” during deployment.
>
> > I think the authors are missing a simpler baseline that is fine-tuned using the final reward (i.e., whether you win or not the game) as a reward as was done in previous work [4].
>
> We computed this (below). However, it’s very important to note that this is not a baseline, but an alternative that **uses privileged information that our approach does not have access to**, because our approach has no information about task success, but only the human utterances that the model was exposed to. This is a really important design decision, which is critical for applicability to real-life scenarios.
>
> We simulate Interaction-Oracle learning on B-SUP datasets. We report the exact match rate on an offline validation set of 344 turns from human-human games (not exactly in the distribution of human-bot games, but this is informative).
>
> | Round   | B-SUP | Interaction-Oracle
> | ----------- | ---------  | ---------
> | 1 | 43.3%    | 43.8%
> | 2 | 45.9%    | 43.3%
> | 3 | 49.1%    | 46.2%
> | 4 | 48.5%    | 47.3%
> | 5 | 53.1%    | 47.1%
> | 6 | 52.0%    | 47.7%
>
> Providing clean and granular progress rewards achieves higher validation rates except for the first round (even though B-SUP has no idea about task success). The model picks up intermediate mistakes in the process by filtering solely based on the final task success. We also overestimate the performance of Interaction-Oracle with offline simulations here, because they were trained on interactions collected from a better model (potential model distillation) – so please consider these numbers an overestimate.
>
> To be continued in Part 2.

---

> > ### Author Response · Authors · 2024-11-20
> > **Initial Response Part 2**
> >
> > > The related work cites some interesting work related to using AI-generated feedback for improvement. However, the authors do not provide a baseline where this is explored for this game. For instance, the method proposed by Yuan et al 2024.
> >
> > We do not use AI-generated feedback. This is an important aspect of our approach, because it means we do not rely on the ability of the model (or another model) to judge performance. The signal comes from followup human utterances. It’s decoded by the LLM. AI-generated feedback uses a model to judge the model output (i.e., actions). We don’t do that. We will highlight this difference in the related work, and note this separate thread of research.
> >
> > ## Response to Additional Suggestions
> >
> > We will review the manuscript to revise as needed. In the meantime, here are some answers:
> >
> > - Figure 4 Center shows the trend in the average number of dialogue turns over different rounds.
> > - We use Empirica to pair workers. Pairing is completely random – same for pairing humans with each other and our model. This is to best simulate deployment. We do not make any effort to optimize for successful interactions, for example, because that would introduce biases into our deployment. More details on MTurk setup are in Appendix A3.1.
> > - We did not include the baseline term in REINFORCE implementation, see Eq (2). We opted for vanilla REINFORCE for its simplicity. PPO, for example, would have required a value function, and would likely be a poor fit for our offline setup. However, there’s room to try other techniques in future work. KTO represents a more complex and recent method, and our mixed results show the challenges of deploying it in our setup, highlighting an important area for future work.
> > - Typo is fixed.
> > - We will make a pass to make the use of terms (LLM, MLLMs, etc) consistent.
> > - Figure 1 and feedback type: the thumbs up in Figure 1 suggests positive feedback. We add explanations in Figure 1 caption.
> >
> > ## Response to Raised Questions
> >
> > > It's not clear to me how the authors complete their fine-tuning considering that most of the training regimes are not designed for dialogue data specifically. For instance, if you have a dialogue of 4 turns, do you simply treat this as a single example or do you derive many examples for it? This is an important detail which I don't think is specified in the section that describes the training regime.
> >
> > Please see our response to the relevant point above. Related details are specified in Section 3 on MultiRef.
> >
> > > What kind of REINFORCE implementation did you use? Did you adopt a baseline term? I think it's important to report more detail to aid reproducibility
> >
> > Please see in the additional suggestions above.
> >
> > > Considering that the action space of the model is very limited, have you considered a form of token masking to improve the performance of your algorithms?
> >
> > Models learned the output format well from data alone without token masking. Post-hoc error analysis also revealed that outputs are valid. Errors are genuinely selecting a valid but wrong token. Constrained decoding helped KTO in round 3 a bit to counter the instability of B-KTO, but it wasn’t the model’s only problem (it just selected bad targets). It was not necessary otherwise.
> >
> > Please let us know if you have further questions, and consider raising the overall score if our responses are helpful.

---

> ### Comment · Reviewer_Aocp · 2024-11-25
> **Response**
>
> Thanks for the clarifications; these are really useful indeed. However, I would like to touch upon a point that does not convince me entirely:
>
> > We computed this (below). However, it’s very important to note that this is not a baseline, but an alternative that uses privileged information that our approach does not have access to, because our approach has no information about task success, but only the human utterances that the model was exposed to. This is a really important design decision, which is critical for applicability to real-life scenarios.
>
> I wouldn't consider this as privileged information. If you're playing a game, you always know at the end if you won or lost. So I consider your design choice as an artificial constraint that your model doesn't need to have. So technically using the task success as an additional reward is a totally legit decision which is not accessing *privileged* information.
>
> Happy to reconsider my score once this is clarified further. I also thank the authors for adding the additional experiment.

---

> > ### Author Response · Authors · 2024-11-26
> >
> > Our use of the MultiRef scenario is a laboratory proxy for studying real-world human-model interactions. ReSpect is designed for such scenarios, where there is no game-completion signal for success. Imagine interactions between a human user and ChatGPT. There is no easily accessible ground truth task success or “game” points for answering how to plan a family outing or touching up a paragraph (without external annotation – which we do not assume). Instead, we propose to learn from implicit rewards humans emitted during the interaction. This gives us granular feedback and enables us to learn from even incomplete interactions, and without annotation effort. MultiRef allows us to study this in an academic setting because we can deploy multiple rounds, collect live interactions, and evaluate the policy exactly.

---

> > > ### Comment · Reviewer_Aocp · 2024-11-27
> > >
> > > Many thanks for this. I understand the overall goal a bit better now. I feel like this entire narrative should be clarified in the paper to make sure that the reader understands:
> > >
> > > 1. I why you selected this dataset as your reference and not any of the other that I've listed above. I believe that having this systematically reported in a table would be really valuable for a reader;
> > > 2. make sure that you justify the notion of learning continually in interaction and why this task is so well suited for it (in addition to the important point that you ideally don't need a task success measure);
> > > 3.  clarify how you prevent overfitting and you assess task generalisation once your agent is deployed. Potentially reporting in the limitations section how this approach can generalise to the other tasks that I've reported might be useful for discussing future work.
> > >
> > > Overall, I believe that this paper represents a good contribution however, as highlighted by other reviewers, requires some edits to make sure that the narrative flows nicely and the goals and contributions are clear. As a result of my current perception of the paper, I've increased the score for contribution to the field.

---

> > > > ### Author Response · Authors · 2024-11-27
> > > >
> > > > These are relatively minor edits, and we agree they will help the paper. Thanksgiving is upon us now (and most authors are offline), but we will incorporate these changes. The PDF is not updateable after today. We will include 1+2 in your list in a section that compares MultiRef to the datasets you listed, and do a pass over the intro to strengthen the justification. We will add an explicit discussion of limitations (something that got dropped in the last minute as we going for submission), among other things discussing generalization.
> > > >
> > > > Given that we have converged on the needed edits and clarified all issues, we hope you will consider updating your score. I think that we both agree that this is an important contribution, and a perspective on learning that will benefit the ICLR program.
> > > >
> > > > Thanks (and happy holiday, if you are celebrating)!

---

### Official Review · Reviewer_Gzkj · 2024-11-03

**Soundness:** 3
**Presentation:** 3
**Contribution:** 2
**Rating:** 5
**Confidence:** 4

**Summary:**

- The paper presents RESPECT, a framework for LLMs to learn from implicit user feedback in multi-turn interactions. Rather than relying on external annotations, RESPECT enables models to retrospectively analyze past interactions and learn from cues like rephrased requests, signs of user approval, or frustration.
- This approach is applied in MULTIREF, a new multi-turn reference game where users instruct the model to select abstract shapes (tangrams), and the model gradually improves its accuracy based on decoded feedback signals.
- The study compares three learning strategies: supervised learning, REINFORCE, and KTO, finding that models using only positive feedback perform best.

**Strengths:**

- The use of continual learning in the RESPECT framework demonstrates strong potential for developing LLMs that improve continuously from real-world interactions.
- The retrospective aspect of RESPECT is particularly compelling, as it enables models to learn from user corrective feedbacks.

**Weaknesses:**

- The experiments are confined to the MULTIREF scenario with abstract tangram shapes. This limited scope raises questions about the generalizability of RESPECT to other domains. Applying RESPECT to diverse settings, such as conversational agents could demonstrate its robustness and adaptability across a broader range of applications, particularly those involving complex language or high-stakes interactions.
- There's a risk that the model might overfit to specific patterns of implicit feedback rather than truly improving at the task.
- The paper does not compare RESPECT to other established methods for learning from implicit feedback or continual learning. Without such comparisons, it's difficult to assess the relative merits of this approach For example, methods in RLHF using preference modeling or utility maximization strategies  could serve as useful baselines.
- The feedback decoder relies on the model's ability to interpret implicit signals correctly. However, there's no guarantee that the model's interpretation aligns with the human's intended feedback. The paper would benefit from a more thorough analysis of cases where feedback may be misinterpreted and how this affects learning.
- While the paper shows improvement over six rounds for B-SUP, this may not be sufficient to fully understand long-term learning dynamics. The observed plateau and temporary decrease in performance warrant further investigation. Extended experiments over more rounds could provide insights into whether the approach continues to improve or stabilizes at a certain level.

**Questions:**

- How well do you expect RESPECT to generalize to other domains or tasks beyond MultiRef? Have you tested it in any other scenarios?
- Have you considered ways to mitigate the impact of feedback misinterpretation on learning?
- Have you considered any potential ethical implications of learning from implicit human feedback, such as privacy concerns?
- The paper mentions that negative feedback signals are generally underutilized due to challenges in integrating them effectively. Would a more nuanced approach to weighting or categorizing negative feedback improve the model’s performance? eg some negative feedback could carry more importance than others. For instance, if the user strongly corrects an action (e.g., "No, that's completely wrong"), this feedback could be weighted more heavily than a milder form of dissatisfaction (e.g., "Not quite right"). Assigning different weights would allow the model to learn more from severe mistakes than minor ones.
- What are the computational costs of implementing RESPECT, especially the retrospective analysis of past interactions?
- In Figure 3, there appears to be a formatting issue or typo. It says "positive or negative positive, neutral, or negative feedback," which seems confusing. Likely, this is unintended and should read either "positive, neutral, or negative feedback" or "positive or negative feedback".

---

> ### Author Response · Authors · 2024-11-20
> **Initial Response Part 1**
>
> We thank the reviewer for their helpful comments. We address the concerns below, and following this discussion, we will upload the edits (the changes are not yet in the PDF).
>
> ## Response to Raised Weaknesses
>
> > The experiments are confined to the MULTIREF scenario with abstract tangram shapes.
>
> > How well do you expect RESPECT to generalize to other domains or tasks beyond MultiRef? Have you tested it in any other scenarios? (Question 1)
>
> It’s important to consider the research trade-offs and implications of asking for more scenarios. We deployed multiple variants of our approach to interact with humans over thousands of interactions. This is the only way to study such methods. However, it’s also very expensive and time consuming. Yes, there may be insights to be gained by more scenarios, but would this kind of research be feasible in an academic setting? Where budgets are limited and teams are tiny. The implications of raising the costs even higher is pushing academic research out. The problem we study is important. As a research community, we must strike the right balance to make progress. Our research contributes insights and methods to drive this important direction forward.
>
> As we discuss in the related work, [Don-Yehiya et al.](https://arxiv.org/abs/2407.10944) showed similar signals exist in scale (unlike us, they didn’t experiment with continual learning and a model bootstrapping itself). We believe their conclusion complements our work, showing that general scenarios elicit similar signals.
>
>
> > There's a risk that the model might overfit to specific patterns of implicit feedback rather than truly improving at the task.
>
> This is an empirical question. Our experiments empirically show consistent improvement in task performance over time, showing the model is improving at the task. We also extensively evaluate the feedback decoding. It’s not perfect, but very effective. There are few false negatives classified by the LLM feedback decoder (see the confusion matrix – Figure 6). False negatives are higher though, suggesting there is room to improve further (only making the approach more effective). The task success rate improved significantly (by 51%) though, even with this imperfect reward decoder.
>
> > The paper does not compare RESPECT to other established methods for learning from implicit feedback or continual learning.
>
> We are not certain what other implicit feedback methods the reviewer is referring to. We are happy to review any specific approach.
>
> It’s important to note that continual learning is at times used to describe scenarios where models are adapted to new tasks. We use it in the sense of improving a model on its original task over time. So methods for domain adaptation are not relevant.
>
> With regard to RLHF-ish methods: we experimented with KTO (a utility maximization method), and presented extensive deployment results with it. General RLHF methods and DPO-like methods are not applicable to our domain because they rely on paired preference data, but we have feedback on single outputs. This is why we used KTO, which allows using single-example feedback.
>
> > The feedback decoder relies on the model's ability to interpret implicit signals correctly. However, there's no guarantee that the model's interpretation aligns with the human's intended feedback. The paper would benefit from a more thorough analysis of cases where feedback may be misinterpreted and how this affects learning.
>
> We conduct extensive human evaluation to quantitatively assess how well the interpreted signals align with human intended feedback and their impact on learning in Section 6. To summarize:
>
> - The LLM feedback decoder yields negligible false positives (<2%), and consistently reaches >60% accuracy considering all classes (diagonals in confusion matrix). This also suggests the training data for supervised systems are fairly clean (because they rely on positive signals only).
> - The false negative rates are higher than false positives (around 15% for binary decoders, and 20% for ternary decoders throughout rounds). As a result, systems using ternary decoders recovered fewer data points as we discard predicted neutrals.
> - To assess the impact of the quality of feedback decoders on learning, we adapt both binary decoders and ternary decoders for each optimization strategy. The deployment results (Figure 4) reveal that binary decoders have a slight edge over ternary decoders with B-SUP v.s. T-SUP, suggesting that more data is perhaps more important than cleaner data for supervised learning at least. The benefit is negligible for learning with RL systems.
> - The human’s intended feedback is fairly unambiguous, supported by 93% agreement rate among three annotators.
>
> To be continued in Part 2.

---

> > ### Author Response · Authors · 2024-11-20
> > **Initial Response Part 2**
> >
> > > While the paper shows improvement over six rounds for B-SUP, this may not be sufficient to fully understand long-term learning dynamics. The observed plateau and temporary decrease in performance warrant further investigation. Extended experiments over more rounds could provide insights into whether the approach continues to improve or stabilizes at a certain level.
> >
> > It’s important to consider the cost of each round of experiments, and the balance between the cost and insights. Each round costs $2000 for all six systems. Our experiments already show >50% task performance improvement, which tells us there is dramatic learning over time. There’s always the possibility that the next round will show even high performance, or a plateau. But this doesn’t change the fundamental answer to the research questions.
> >
> > We discussed the plateau in Line 374 and additional experiments in Appendix D. To summarize: we observe the performance was capped by the LoRA adapters and conducted additional experiments with more expressive adapters, which recovered monotonous improvement in task success rate. We suspect the overall slowdown is because the optimal hyper parameters might have changed along continual learning (i.e., as the amount of data increased and/or the distribution shifted).
> >
> > ## Response to Raised Questions
> >
> > > Have you considered ways to mitigate the impact of feedback misinterpretation on learning?
> >
> > See the discussion of the feedback decoder above. In addition, there’s more room to develop better reward decoders. For example, given some investment in data annotation, one could fine-tune a separate LLM to decode the feedback (we focused on showing the LLM bootstrapping from its own interactions without such annotations, but this is a possible direction).
> >
> > > Have you considered any potential ethical implications of learning from implicit human feedback, such as privacy concerns?
> >
> > Deployment of our approach suffers the same risks as approaches that fine-tune on interaction data. This is something that companies do all the time (with annotations on the data), and they deploy different techniques to filter out the data to not include sensitive/private information. We will note this in the paper.
> >
> > > The paper mentions that negative feedback signals are generally underutilized due to challenges in integrating them effectively. Would a more nuanced approach to weighting or categorizing negative feedback improve the model’s performance? ...
> >
> > This is a great idea for future work. We will note it in the discussion. This is the first work in this space, so we opted for simplicity.
> >
> > > What are the computational costs of implementing RESPECT, especially the retrospective analysis of past interactions?
> >
> > The computational cost of retrospective analysis is minimal compared to retraining policy models. Here, prompting-based retrospection takes less than 3 minutes on 300 interactions (around 2400 actions) on a single A6000 GPU, without inference optimization.
> >
> > > In Figure 3, there appears to be a formatting issue or typo. It says "positive or negative positive, neutral, or negative feedback," which seems confusing. Likely, this is unintended and should read either "positive, neutral, or negative feedback" or "positive or negative feedback".
> >
> > This is not a typo. Figure 3 presents prompt templates for both the binary and ternary feedback decoders by color coding. We will make sure it’s clearer in the caption.
> >
> >
> > Please let us know if you have further questions, and consider raising the overall score if our responses are helpful.

---

> > > ### Comment · Reviewer_Gzkj · 2024-11-26
> > >
> > > Thank you for your detailed responses and for addressing my comments. While I understand the constraints of conducting research in academic settings, I still remain concerned about the risk of overfitting and scalability.
> > > -  The model's continual fine-tuning over limited data raises concerns about overfitting to patterns specific to the MULTIREF scenario. Although the authors argue that consistent task improvements demonstrate generalization, these improvements may stem from the model being repeatedly evaluated on a narrow and static task. This does not necessarily indicate true learning or generalization, as the model's gains could reflect a form of memorization or adaptation to the MULTIREF setup rather than the development of robust capabilities transferable to other domains. Validation in diverse settings is critical to confirm the broader applicability of RESPECT.
> > > - While I acknowledge the cost and logistical challenges of testing RESPECT across varied scenarios, scalability is a critical requirement for any framework aiming for real-world deployment. The reliance on high-quality human interaction data makes scaling difficult, particularly in diverse or open-ended domains where such data may be sparse, noisy, or expensive to obtain. This limitation could hinder broader adoption, especially for applications requiring diverse language understanding or complex real-world reasoning.
> > > - The observed plateau in task performance raises significant questions about the framework's ability to sustain improvement over extended rounds. While I appreciate the authors' explanation regarding hyperparameter expressivity, this further underscores the need for adaptive learning techniques that can handle evolving data distributions and maintain progress. Additionally, the continual fine-tuning approach introduces risks of catastrophic forgetting or confusion in learning. As new rounds overwrite prior knowledge, the model may improve on the immediate task while losing general capabilities or robustness. This is a fundamental concern in continual learning setups and warrants further investigation to ensure meaningful and sustained learning over time.

---

> > > > ### Author Response · Authors · 2024-11-27
> > > > **Response Part 1**
> > > >
> > > > Thank you for engaging. We respond to your concerns individually below. We swapped the order because some explanation from bullet point 2 can help with bullet point 1.
> > > >
> > > > > Scalability is a critical requirement for any framework aiming for real-world deployment. The reliance on high-quality human interaction data makes scaling difficult, particularly in diverse or open-ended domains where such data may be sparse, noisy, or expensive to obtain….
> > > >
> > > > On scaling data: We agree that scalability is important. Scalability is actually the *strength* of learning from human-model deployment data. Key to our approach is that it relies on the interaction the system has with users during its deployment, as it completes tasks. So these interactions are already taking place. Our approach adds no overhead in terms of interaction or annotation on top of regular interactions with users. This is the most important property of our work: the signal arises naturally from the interactions the system already has with human users. This is different from contemporary RLHF methods, which rely on getting post-hoc preference annotations from third-party annotators. **This is why our approach is a fundamental improvement on current training recipes (e.g., RLHF), and this is why it naturally scales.** As people use the system, we get more signals, and the system autonomously learns from them – without any annotation effort.
> > > >
> > > > On high-quality data: [Don-Yehiya et al.](https://arxiv.org/abs/2407.10944) shows that similar learning signals exist in a large-scale real-world dataset [LMSYS-CHAT-1M](https://arxiv.org/abs/2309.11998) which covers general diverse domains. As we discuss in the related work section, their work complements our focus on continual learning, showing the prevalence of these signals in the interactions people have with existing LLMs. So, the signal is already out there, and it’s already at a very large scale.  Together we think our research strengthens each other and provides justifications for further research in this novel and promising field and broader adaptation.
> > > >
> > > >
> > > > > These improvements may stem from the model being repeatedly evaluated on a narrow and static task. … Validation in diverse settings is critical to confirm the broader applicability of RESPECT.
> > > >
> > > > We agree a diversity of tasks is a natural and important next step. Please see above how the breadth of applicability is strengthened from concurrent work by [Don-Yehiya et al.](https://arxiv.org/abs/2407.10944). We contribute to the long-term applicability with non-stationary data distribution, and the potential of bootstrapping, ruling out concerns for distilling by learning from interactions of stronger models.
> > > >
> > > > It’s important to note that although the domain is scoped, it doesn’t mean no learning is happening. It’s very common to fine-tune models on specific domains and tasks, and this kind of process is generally viewed as true learning.
> > > >
> > > > Scoping down the task is necessary to study the long term effects of human-in-the-loop scenarios in an academic lab. We did not choose a general task with free-form text generation precisely due to the amount of data it can take per round to see an improvement, especially when we are interested in the long-term dynamics that demands multiple rounds of human-model deployment. MultiRef enables us to collect 330 live human-model interactions per round, and see a difference. Is there overfitting to this task? Yes, of course. We certainly do not expect our best MultiRef policy to answer a grade school math problem. The point of this work is not to instill tangram reasoning into IDEFICS2 and deliver model checkpoints, but to demonstrate the feasibility and effectiveness of bootstrapping new skills in LLMs from interactions in deployment, annotation free.
> > > >
> > > > What’s the alternative? Collecting 1 million conversations (like LMSYS-CHAT-1M) for generality? That’s collecting 24 such large-scale datasets by the end of our continual learning (not even considering pilot runs). We would love to see if ReSpect works there, but an experiment at that scale perhaps is only possible in industry. Even in industry smaller scale proofs of concepts are mandatory to secure sponsorships for broad deployments, and that is our work: showing it is possible in the long run to bootstrap from interactions alone. Our work does just this: enables the next step of scaling up.
> > > >
> > > > To be continued in Part 2

---

> > > > > ### Author Response · Authors · 2024-11-27
> > > > > **Response Part 2**
> > > > >
> > > > > > The observed plateau in task performance raises significant questions about the framework's ability to sustain improvement over extended rounds. … This further underscores the need for adaptive learning techniques that can handle evolving data distributions and maintain progress.
> > > > >
> > > > > We agree with the importance of addressing catastrophic forgetting and new adaptive learning techniques. In fact, we will be exhilarated to see new breakthroughs in that area. However, that is simply not the research question of this work: we *do not* propose a general continual learning adaptation technique that solves or aims to solve catastrophic forgetting once and for all. We identify an underappreciated learning signal (implicit human feedback from interactions themselves), and what long-term dynamics it creates when we learn from rollouts of shifting distributions from its own. Continual learning is a lens through which we study the impact of bootstrapping by training on interactions from deployment, and to mimic the develop-deploy cycles of conversation models in production, but continual learning itself is not a research target for its own merit in this work.
> > > > >
> > > > > As a work that *implements (not studies)* continual learning, we took measures to avoid catastrophic forgetting in the face of data distribution shifts (e.g. better task performance, reduced vocabulary sizes, shorter utterances) via data accumulation and rehearsal. To clarify, we did not do “continual fine-tuning”. For each round $\rho$, we load the same checkpoint from IDEFICS2-8b and train on all data so far $D_{\le \rho}$ for supervised and RL systems. Training from scratch is a common technique to overcome the loss of model elasticity in continual learning setups (exactly the problem as you pointed out). This is similar to [Kojima et al. 2021](https://arxiv.org/abs/2108.04812) and [Suhr and Artzi 2023](https://arxiv.org/abs/2212.09710). Empirically these techniques also worked well in MultiRef: we observe strong improvement of an absolute 51% after 6 rounds. Addressing catastrophic forgetting in continual learning is an important research topic, but tangential to our contribution on learning from implicit feedback signals.
> > > > >
> > > > > Lastly, regarding the plateau, it is not due to catastrophic forgetting, because of the aforementioned guardrails, and the fact that we focus on improving one single hard task. We have discussed extensively why it occurred (model expressivity, hyperparameters for continual learning), what helped fix it partially (better LoRA adapters), pointed to future directions such as learning a feedback decoder to uncover more subtle feedback (more nuanced decoder as you pointed out previously). All of these insights, analysis, and takeaways are only made possible *after* this research, i.e., deploying an imperfect framework, for 6 rounds, collecting thousands of human-model interactions, witnessing a significant improvement from 31% to 82%. The insights and lessons from this project are a novel contribution to the community in its own right.
> > > > > What does it take to achieve the perfect 100%? There is always a point of diminishing returns in a research project and we have offered all the insights for future work in the discussion section. We will leave the last 18% as an exercise for thoughtful readers :) Regardless the outcome of new endeavours, they won't change our answer to the research question, that we *can already* effectively learn from interactions in deployments, through multiple rounds and without a single expert annotation.
> > > > >
> > > > > Again thank you for engaging. Let us know if we have clarified why overfitting to MultiRef is from an economical experiment design, and that data scalability is actually a strength of learning from human-model interactions in deployment. Please consider raising the overall score if you find our answers helpful.

---

### Official Review · Reviewer_XxNx · 2024-11-04

**Soundness:** 3
**Presentation:** 3
**Contribution:** 2
**Rating:** 6
**Confidence:** 3

**Summary:**

The paper proposed a method to train a model with implicit human-in-the-loop feedback for a referential game. The proposed method first translate implicit human natural language feedback and quantize them into positive, (neutral), and negative labels, and then use the feedback to fine-tune the language model for decision making.

The experiment is situated in a referential game, where human is serving as a speaker to describe a subset of tangrams, and the model is serving as the listener to pick out the objects the human was describing.

The paper experimented with three learning methods: supervised learning, REINFORCE, and KTO. The models were initialized with pre-trained IDEFICS2-8B weights and fine-tuned with LoRA. Each model setting was fine-tuned with 0/1/2/3 rounds (B-SUP for 6 rounds), before being deployed in the online setting to have human-bot evaluation.

The paper observed that supervised learning method with binary quantization provided best performance, and that the feedback decoder's performance is relative stable across rounds and is consistent with human evaluation. The paper also observed that the human language is getting simpler with smaller vocabulary size and reduced utterance length across the rounds.

**Strengths:**

1. The paper proposed a learning method, RESPECT that utilizes implicit human-in-the-loop feedback for explicit action improvement
2. The paper experimented with 3 learning methods: supervised learning, REINFORCE, and KTO
3. The paper conducted thorough experiments in a multimodal referential game
4. The paper conducted pre-training as well as online testing for iterative model improvement and evaluation
5. The paper is very well structured and well-written. The paper analyzed in detail about learning strategy tradeoffs, feedback label selections, feedback decoder evaluation, and language analysis.

**Weaknesses:**

The paper wishes to highlight the contribution on 'continual learning' and model's iterative improvement with human's online feedback, but the actual experiments conducted is slightly misleading. The authors were careful to distinguish the differences between 'round' and 'turn.

- In the setup, each 'round' includes multiple 'turns' of interactions between a human and the bot.
- The model is retrained after each 'round', with the history of all previous 'rounds'
- After fine-tuning at the end of each 'round', the model is fixed and deployed for evaluation

The main difference between proposed method versus the classic fine-tuning is the increasing context length during each fine-tuning round. It is unclear what the intended benefit is for the increasing context history?
For example:
- Interaction history could help personalize the message or have a better understanding the counter party's message, if the bot was interacting with the exact same human. It was unclear if the bots were interacting with the same human users across different rounds.
- Interaction history could help the bot understand the task goal, through multiple rounds of probing and try-and-error (like RL), only if the bot was not briefed on what the task goal (referential game) was. According to the experiment setup (Figure 1, 2, 3), the model seems to have prior knowledge of the exact task goal.

Beyond the examples illustrated above, the improved performance demonstrated in the paper might just be a result of fine-tuning with more data, and the expensive online evaluation among different turns showcased model's intermediate checkpoint performance.

Nevertheless, the paper proposed a new method that turns implicit human feedback into explicit rewards that could help improve model's performance. It is the 'Continual Learning' aspects that lacks sufficient support.

**Questions:**

Ln 425-426: According to Section 4.2, RL includes both positive and negative rewards. What might be the reasons that including extra rewards would 'encourage a uniform distribution'?

---

> ### Author Response · Authors · 2024-11-19
> **Clarification on Continual Learning and answer questions**
>
> We thank the reviewer for their helpful comments. We address the concerns below, and following this discussion we will edit the paper to make sure the continual learning aspect comes through clearly (the changes are not yet in the PDF).
>
> ## Response to Raised Weaknesses
>
> We define **continual learning** as the model improving over time on its task through interaction with human users. Our data collection and evaluation are both integral parts of our deployment. At each round, we deploy the current model to interact with humans. Each system has its own set of interactions, and each system is trained on the interactions that **it** had with the human users, so its own performance improvement influences its continual learning. In this process, the model continually improves over time from the signals obtained from its own interactions with humans.
>
> In this sense, whether the task has multiple turns (like in our case) or not is not a factor of the setup. You could think of the model learning within single interactions (improving from earlier in the interaction to later), but this is not within the scope of our work. Our work is also not about adaptation, because we don’t maintain user-specific models. This is possible with our approach, but not within the scope of our experiments.
>
> The term continual learning is used broadly, for example for domain adaptation. We will add a clarification very early in the paper to how we use it. We apologize for the confusion. Please let us know if there is anything we can add to clarify the setup.
>
> There is also no direct relation between the progression of rounds and the content length (context in the sense of the LLM prompt, see Figure 11). The context is longer in later turns in the same interaction because the history is longer (more past turns), but this is true both in early and later rounds of the continual learning setup.
>
> Why is continual learning so important for our study of implicit conversation signals? There are complex dynamics between the signals we use and how the system evolves by learning from them. For example, it could be the case that as the model gets better at the task, our feedback decoding process becomes worse. We show this is not the case, and that the signals are present throughout the system’s lifetime (even as it improves) and our decoding approach remains effective and stable. Our setup also mimics the development-deploy cycles in practice, where the data collected in production often correspond to multiple prior model checkpoints.
>
> ## Response to Raised Questions
>
> > Ln 425-426: According to Section 4.2, RL includes both positive and negative rewards. What might be the reasons that including extra rewards would 'encourage a uniform distribution'?
>
> If we understand correctly, you are asking about why the inclusion of negative rewards “encourages a uniform distribution”. This was observed by [Kojima et al. 2021](https://arxiv.org/abs/2108.04812). Essentially, by pushing down the probability of one output (the output with a negative reward), the model increases the probability of **all** other outputs. Because it’s a very large output space, this essentially pushes towards a uniform distribution. This is similar to unlearning objectives, which aim for the same (i.e., uniform distribution).
>
>
> Please let us know if any questions remain, and consider raising the overall score if our responses are helpful.

---

### Official Review · Reviewer_aUKE · 2024-11-04

**Soundness:** 3
**Presentation:** 3
**Contribution:** 3
**Rating:** 3
**Confidence:** 4

**Summary:**

The author proposes a novel framework, Retrospective learning from past interactions (RESPECT), for improving the LLMs based on signals from past interactions via retrospection. They also contributed with a task, Multi-turn Grounded Interaction Scenario (MULTIREF), a conversational interaction scenario where two partners, a speaker and a listener, coordinate on the selection of a set of items. They further fine-tuned an LLM with different optimization techniques with the data collected from the proposed task and framework.

**Strengths:**

- The idea of learning from past mistakes is really interesting and the proposed framework, RESPECT, doesn't depend on the optimization strategy. As highlighted in the paper, this framework can be used with various optimization strategy (Supervised Learning, Reinforcement Learning, Utility Maximization).
- The paper also contributed with a new task, MULTIREF, which will be very useful for the future development of this domain.

**Weaknesses:**

**Major:**
- **Excessive use of training data:** The proposed method relies heavily on data. The model is fine-tuned at each step with all the interaction data acquired from past steps. Now, although the authors mention that they are taking measures to avoid overfitting (lines 246-248), this much repeated data usage would eventually result in overfitting.
- **Lack of metric evaluation:** Although the authors showcases various observations and results through plots and confusion matrix, they lack the tables for comparing different metics. Having those results will significantly boost the paper quality.
- **Lack of generalizability of proposed method:**
  - **Across different LLMs:** The proposed framework is only tested on one LLM (IDEFICS2-8B). Now, although this framework can be applied over other LLMs, it is unclear whether it will boost their performance or not. One reason authors might have seen such improvement is because the tested LLM is bad at that particular task. If we have a very good LLM then this framework might not help much as we will have less interaction data to fine-tune model on.
  - **Across different tasks:** Another interesting extension of the proposed method can be over different tasks. Currently authors have only tested over a particular task but it would be interesting to see if it can be extended over other tasks like summarization (authors have highlighted this as future work in discussion).
- **Scalability:**
  - **Heavy reliance on human feedback:** The proposed framework relies heavily on human feedback. Although the authors have countered the problem of annotating the responses, the problem of getting good interaction data still remains a crucial problem, making it hard to scale.
  - **Cost:** Another issue with scalability is the cost associated with getting quality interaction data. As mentioned by authors, this small experiment took over a month to collect data and costed $11k USD (line 347). This also makes it harder to use the proposed method in real time.

**Minor:**
- Adding information about MULTIREF in the introduction will help in understanding the contribution of the paper.
- Information on LoRA configuration used for fine-tuning is missing.
- Very similar labels are used for control and HH data in figure 4, making it hard to interpret. Maybe changing it with something else will make it more interpretable.
- **Repeated variable:** Authors have repeated the use of variable t. At line 90 it represents time while at line 141 it represents turn and at line 145 it is again used as time.
- **Typo:** Fullstop (.) missing in line 422. "(supervised vs. RL/KTO) Overall" &#8594; "(supervised vs. RL/KTO). Overall"

**Questions:**

1. How do you address overfitting given the extensive reuse of training data at each fine-tuning step?
2. Can you provide comparison of models across different evaluation metrics?
3. Have you tested the framework on other LLMs or tasks to confirm generalizability? How might the framework’s usability vary with stronger LLMs that provides less interaction data?
4. Have you considered other optimization technique like Direct Preference Optimization (DPO) which uses the binary labeled data while fine-tuning the LLM?

---

> ### Author Response · Authors · 2024-11-21
> **Initial Response Part 1**
>
> Thank you for your feedback. We have modified our manuscript to incorporate it (uploading soon following this discussion). Regarding the concerns you raised:
>
> ## Response to Raised Weaknesses
>
> > Excessive use of training data: The proposed method relies heavily on data. The model is fine-tuned at each step with all the interaction data acquired from past steps. Now, although the authors mention that they are taking measures to avoid overfitting (lines 246-248), this much repeated data usage would eventually result in overfitting.
>
> We train from **scratch** in every round (6 rounds in total – not every interaction step) for most systems. Training from scratch means: at every round, we fine-tune the initial IDEFICS2-8B parameters. We are not using the data to optimize again and again the same parameters. The measures we took to avoid overfitting are standard, just like any fine-tuning process, and the risk of overfitting is not different from any supervised learning scenario. Our evaluation is on unseen games (every round brings completely new games), showing strong generalization and improvement.
>
> > Lack of metric evaluation: Although the authors showcases various observations and results through plots and confusion matrix, they lack the tables for comparing different metics. Having those results will significantly boost the paper quality.
>
> We will add 8 tables in Appendix E accompanying Figure 4,5,6. Figures 4,5,6 illustrate the trends. We included specific numbers in the Results section with analysis. The confusion matrix is already labeled with numbers so we will highlight the key observations in the caption as well.
>
> > Lack of generalizability ... across different LLMs: The proposed framework is only tested on one LLM (IDEFICS2-8B). Now, although this framework can be applied over other LLMs, it is unclear whether it will boost their performance or not. One reason authors might have seen such improvement is because the tested LLM is bad at that particular task. If we have a very good LLM then this framework might not help much as we will have less interaction data to fine-tune model on.
>
> Our goal is to show implicit conversational signals can drive learning. These signals exist beyond our setup (as we discuss in related work). The poor initial performance is a feature of the experimental design – it allows us to show an effect. Other LLMs might be good at the task, but they will be bad at other tasks, and our approach will allow them to improve over time. Our research question is: Can implicit signals from interactions be leveraged and sustain continuous improvements in deployment? To show its effectiveness, we carefully selected the multi-turn reference scenario from CogSci and tangrams that are unfamiliar to modern pretrained vision LMs (shown in Ji et al.), such that (a) the scenario can be deployed in an academic lab and (b) progress is measurable with a reasonable amount of data. This methodology leads to our evaluation with one model and one task.
>
> Of course, it is always useful to try different models, but it is also important to consider the costs of such experiments – with humans in the loop. What are the implications of demanding many LLMs and many tasks with such studies? It will simply be impossible to run such research in academic labs. Even industry labs are not likely to just put millions of dollars into a speculative new approach without a proof of concept. This is what our research is.
>
> > Lack of generalizability ... across different tasks: Another interesting extension of the proposed method can be over different tasks. Currently authors have only tested over a particular task but it would be interesting to see if it can be extended over other tasks like summarization (authors have highlighted this as future work in discussion).
>
> We agree with the reviewer on experimenting with different scenarios. As we discussed in our related work, a concurrent work by [Don-Yehiya et al.](https://arxiv.org/abs/2407.10944) showed these signals exist in lmsys-chat-1m. So this tells us that these signals are general. We show how they can be used for an LLM to improve its own behavior. Similar to trying different models, more tasks have extreme engineering and experimental implications – this simply makes this kind of work impossible, even though it’s an important and impactful research avenue.
>
> To be continued in Part 2.

---

> > ### Author Response · Authors · 2024-11-21
> > **Initial Response Part 2**
> >
> > > Scalability: Heavy reliance on human feedback: The proposed framework relies heavily on human feedback. Although the authors have countered the problem of annotating the responses, the problem of getting good interaction data still remains a crucial problem, making it hard to scale.
> >
> > Key to our approach is that it relies on the interaction the system has with users during its deployment, as it completes tasks. So these interactions are *already taking place*. Our approach adds no overhead in terms of interaction or annotation on top of regular interactions with users. This is the most important property of our work: the signal arises naturally from the interactions the system already has with human users. This is different from contemporary RLHF methods, which rely on getting post-hoc preference annotations from third-party annotators. This is why our approach naturally scales. As people use the system, we get more signals, and the system autonomously learns from them – without any annotation effort. The annotations we conduct are for evaluation purposes only, and are *not* part of the learning approach.
> >
> > > Cost: Another issue with scalability is the cost associated with getting quality interaction data. As mentioned by authors, this small experiment took over a month to collect data and costed $11k USD (line 347). This also makes it harder to use the proposed method in real time.
> >
> > The costs we list are for our experiments. We need interactions with users, so we use MTurk. This is what causes the high experimental costs. A company that deploys our approach will not suffer these costs, because the interaction will arise naturally from its models being used (i.e., in a product like ChatGPT). The reason the process took a month is because we worked with a group of workers on MTurk, and their availability was limited (also, training took time because we don’t have very strong GPUs). This time scale is completely immaterial to deploying our approach in a real environment – it’s completely a *byproduct of an academic experiment*.
> >
> > ## Response to Minor Issues Raised
> >
> > - We revised the introduction of MultiRef in the intro to give a better understanding of its main principles.
> > - LoRA implementation details are in Appendix B.3 and Appendix D.
> > - We update Figure 4 with different markers and line styles for HH and Control.
> > - Repeated variables (t): time and turn indeed refer to the same thing; we have updated to use “turn” consistently throughout.
> > - We fixed the typo.
> >
> > ## Response to Questions Raised
> >
> > > How do you address overfitting given the extensive reuse of training data at each fine-tuning step?
> >
> > We clarify why this is not an issue above.
> >
> > > Can you provide comparison of models across different evaluation metrics?
> >
> > We are adding them. Please see the detailed answer above.
> >
> > > Have you tested the framework on other LLMs or tasks to confirm generalizability? How might the framework’s usability vary with stronger LLMs that provides less interaction data?
> >
> > Please see the discussion above regarding the costs and feasibility within academic constraints. Furthermore, IDEFICS2-8B was by far the best open-weight multimodal LLM available when we conducted the experiments. Of course, now there are better models (it’s a fast paced space), with which we just expect faster and even more efficient learning.
> >
> > > Have you considered other optimization technique like Direct Preference Optimization (DPO) which uses the binary labeled data while fine-tuning the LLM?
> >
> > DPO required paired preferences. We don’t have this kind of data, and we can’t get it without additional annotation or overhead on the interaction. This is why we experiment with KTO, a related method that only requires the kind of single-example feedback we can decode (i.e., unpaired preference).
> >
> > We will update the new PDF soon but want to kick-start an engaging discussion first. Please let us know if any questions remain, and consider raising the overall score if our responses are helpful.

---

> ### Comment · Reviewer_aUKE · 2024-11-26
>
> Thank you for your response and addressing most of my queries. I would like the authors to further clarify on the framework and its usability with respect to their response.
>
> > We train from scratch in every round (6 rounds in total – not every interaction step) for most systems. Training from scratch means: at every round, we fine-tune the initial IDEFICS2-8B parameters. We are not using the data to optimize again and again the same parameters. The measures we took to avoid overfitting are standard, just like any fine-tuning process, and the risk of overfitting is not different from any supervised learning scenario. Our evaluation is on unseen games (every round brings completely new games), showing strong generalization and improvement.
>
> I am a bit confused. My initial understanding was the framework finetunes the test LLM model (and not the base LLM model) based on all the data collected from past interactions. This flow has been depicted in figure 1 (on the very first page) of the paper. It clearly shows that $\theta_{p}$ is updated to $\theta_{p+1}$ based on all the aggregated data $D_{p}$. But now the authors claim that this is not the case—at each round, they fine-tune the initial IDEFICS2-8B parameters. This would mean that $\theta_{0}$ is updated to $\theta_{p+1}$. This brings up the question, how is it any different than fine-tuning from scratch, all that is changing is the dataset. The model is not improving on the learned information, all that is happening is model is learning new information. And the reason we see improvement over rounds is because the dataset size is being increased compared to the last round. Maybe I am missing something here and would like authors to clarify my misunderstanding.
>
> > > Have you tested the framework on other LLMs or tasks to confirm generalizability? How might the framework’s usability vary with stronger LLMs that provides less interaction data?
>
> > Please see the discussion above regarding the costs and feasibility within academic constraints. Furthermore, IDEFICS2-8B was by far the best open-weight multimodal LLM available when we conducted the experiments. Of course, now there are better models (it’s a fast paced space), with which we just expect faster and even more efficient learning.
>
> I understand the author's point on cost within academic constraints. My question was more stemmed on the usability of the framework with better multimodal models. As we have seen this trend with LLMs, we can expect better multimodal modals in few years. With those models we might not have enough interaction data for a particular task to fine-tune. Now, the authors says that with better models we can expect faster and efficient learning, but it is based on the fact that model is not performing well (the only way to get interaction data). If it performing well already then we can't expect expect faster and efficient learning. I would like to know authors take on this.

---

> > ### Author Response · Authors · 2024-11-27
> >
> > Thank you for engaging. Below are our follow up to your questions:
> >
> > **Data usage and clarification on Figure 1**
> >
> > This is a good point, and we see why Figure 1 can be confusing. We will fix this. Our intention is that the overarching $\theta_{\rho+1} \leftarrow \theta_{\rho}$ indicates *the model in deployment* is updated from $\theta_{\rho}$ to $\theta_{\rho+1}$, and not referring to the training process.
> >
> > > This brings up the question, how is it any different than fine-tuning from scratch, all that is changing is the dataset. The model is not improving on the learned information, all that is happening is the model is learning new information. And the reason we see improvement over rounds is because the dataset size is being increased compared to the last round. Maybe I am missing something here and would like authors to clarify my misunderstanding.
> >
> > The system improves, because the next model is better. The difference lies exactly in the dataset - it is non-stationary across continual learning rounds. The dataset $D_{\le \rho}$ is *not* collected by simply letting humans interact with the same model for more iterations, yielding homogeneous data points as assumed by static datasets. Instead, $D_{\le \rho}$ grows by including more interactions between humans and **newer/better** models (that were trained on existing interactions). There is a stark distribution shift in data, as illustrated in Figure 4 and linguistic analysis: increased task performance, fewer turns per interactions, reduced vocabulary size, etc.
> >
> > We opted to train from scratch because it simplifies the setup. This is similar to [Kojima et al. 2021](https://arxiv.org/abs/2108.04812) and [Suhr and Artzi 2023](https://arxiv.org/abs/2212.09710). Fine-tuning complicates the setup because of repeated optimization of the model. There is one important exception: KTO. We follow as close as possible to the conventional KTO recipe, so there we fine-tune the same model again and again.
> >
> > Why is continual learning so important for our study of implicit conversation signals? There are complex dynamics between the signals we use and how the system evolves by learning from them. For example, it could be the case that as the model gets better at the task, our feedback decoding process becomes less effective. We show this is not the case, and that the signals are present throughout the system’s lifetime (even as it improves) and our decoding approach remains effective and stable. Our setup also mimics the development-deploy cycles in practice, where the data collected in production often correspond to multiple prior model checkpoints.
> >
> > **Usability of the framework with better multimodal models**
> >
> > Yes, we also expect models to get better. This only strengthens the importance of our approach. The interaction data we use arises from users using the system, and our approach shows an avenue to improve systems in exactly the scenario you raise: when we can’t get a lot of data to fine-tune in advance. What we show is that you can deploy such (M)LLMs, and have them improve from their interactions, even if they start bad.
> >
> > Strong models don’t make our approach redundant, because even if models perform perfectly when released, the world keeps changing, and humans keep changing how they use models. Our approach allows models to learn and adapt continually over their lifetime. Therefore, there will always be room to improve and to continue learning (just like humans do in a changing world 🙂).
> >
> > Of course there is the middle ground where the model is better but still not perfect. Suppose by using IDEFICS3, we started at a MultiRef policy with a task success rate of roughly 50% (approximately IDEFICS2 at round 1) instead of 31% (round 0). Now we learned that our overall framework can support at least 30% improvements in the long run, without any additional annotation! We learned that the feedback decoder will keep working really well despite distribution shifts due to stronger models. We learned that towards later rounds we may need to tune hyperparameters, add model expressivity, increase the number of new interactions added per round, or with a learned feedback decoder to recover more subtle feedback (as we pointed out in discussion). All of these improvements, insights, and future directions on the promising topic of “learning from implicit conversational feedback”, are only revealed *after* this research. That’s the value of our work and we believe sharing them contributes to the knowledge base of this community, even if MultiRef (or any other prototypical task scenarios) becomes trivial for multimodal models eventually in the future.
> >
> > Again thank you for engaging. Do let us know if we clarified on continual learning and addressed your concerns about usability with better multimodal models, and please consider raising the overall score if you find our answers helpful.

---

> > > ### Comment · Reviewer_aUKE · 2024-11-27
> > >
> > > Thank you for your response and further clarification on data usage and utility of framework. Based on our discussion, I would like to keep my scores.
> > >
> > > The current setup does not effectively depict continual learning, which is the essence of the paper. As clarified by the authors, the new dataset is being used to fine-tune the base (original) model ($\theta_{0}$) rather than the model from the previous step ($\theta_{p}$). At each step, the model learns new information instead of building upon past knowledge in a continuous manner. This approach represents isolated learning rather than continual learning. The observed performance improvement is likely due to the increasing dataset size after each round, providing more samples for learning. This raises questions about the paper's core contribution.

---

> > > > ### Author Response · Authors · 2024-11-27
> > > >
> > > > We are sorry, but this is wrong. You follow a narrow definition of continual learning, which misses the whole point of what is continual learning by focusing on a marginal implementation detail (that should be empirically driven, and this is how we decided it). The system is continually learning by interacting with humans, because the data is coming from the current generation of the system interacting with users (this is how the data is collected). Whether we fine-tune the most recent model, or just train with the aggregated data so far from the initial parameters is immaterial to if continual learning takes place or not. Our choice simplifies optimization problems and avoids tuning regularization methods like KL or rehearsal. Your narrow definition is misaligned with a lot of existing work (see the most recent EMNLP best paper award and many other papers).
> > > >
> > > > > The observed performance improvement is likely due to the increasing dataset size
> > > >
> > > > The new data comes from newer generations of the systems interacting with people. It does not appear form thin air or additional annotation. Even if this data is not from a different distribution (which is clearly the case because performance increases over time), the source of the data matters.

---

### Author Response · Authors · 2024-11-21
**General Response**

We thank the reviewers for their comments. We appreciate the reviewers warm words about the work as “the ideas … is really interesting”, the interaction scenario being “very useful for the future development of this domain” and this work “demonstrates strong potential for developing LLMs that improve continuously from real-world interactions”. We fully respond to each review in a thread following their review. Below are answers to critical general questions:

**Why continual learning?**

Continual learning is at times used to describe scenarios where models are adapted to new tasks. We use it in the sense of improving a model on its original task over time. Studying our approach in a continual learning setting allows us to observe if the reward decoding is robust as the system improves and the data distribution shifts. We indeed show the decoding is robust and stable. Studying this kind of dynamics, as we do (with thousands of real-time human interactions), is critical to better understand the methods that learn from feedback.

**Why one model and one task?**

All our experiments are human-in-the-loop, and all model interactions are conducted with humans. This is critical to reflect the true potential of interactions with humans for learning. However, this significantly complicates our studies and makes them very costly. This is a cost that goes beyond the cost of buying compute hardware, which researchers often consider, and is less influenced by just trying another task (if one that fits exists) or model (if an open-weights one exists). Our experiments cost >$11k, and this is without pilot experiments and studies. Such crowdsourcing funds are gone (it’s not hardware you can use again), and are very hard to raise. It’s important to balance utility vs. costs. Our studies provide clear answers to our research questions: multi-turn interaction data contains useful implicit signals, our method can decode it robustly, and it can learn from it rapidly.

In addition, one must also consider the engineering cost of building the scenarios for other tasks. MultiRef is an interactive platform, where humans are paired with each other or with models.

As one considers asking for more models and more tasks, it’s critical to consider what it means for academic research on such problems. If the bar is unnecessarily high for such studies, it simply says one shouldn’t study them in academic settings. This will be a terrible loss for academia, as interactions are increasingly core to how models are used and trained.

**Why not DPO?**

Our scenario doesn’t support DPO, because it does not provide paired preference data, as DPO requires. Our reward decoding process provides a reward for a single output. Paired preferences require two alternatives. This adds overhead to the interaction, which our approach does not. We experimented with KTO though.

---

### Meta-Review · Area_Chair_icSk · 2024-12-11

**Metareview:**

The authors present a way to farm supervision signals from LMs in multiturn interactions and use this as training data to improve the model. The novelty of this approach beyond reward-modeling / RLHF / RLAIF is not very clear. While authors present this as continual learning (which could be a way to distinguish from RLAIF i.e. continuing to learn post deployment), it still requires parameter updates and is trained over the whole dataset each time.

I agree there is something novel here in that humans interacting with the system are not expressly asked to rate the interactions, it's more implicit. But this novelty is not motivated clearly or sufficient for a main conference paper.

**Additional Comments On Reviewer Discussion:**

Significant engagement from reviewers -- sticking points are scalability if human oversight and clarity in scoping the contribution.

---

### Decision · Program_Chairs · 2025-01-22

Reject